# Anomalous water molecular gating from atomic-scale graphene capillaries for precise and ultrafast molecular sieving

Qian Zhang[1,6], Bo Gao[2,6], Ling Zhang[3,6], Xiaopeng Liu[2], Jixiang Cui[2], Yijun Cao[3], Hongbo Zeng [4] ✉, Qun Xu [1,2] ✉, Xinwei Cui [1] ✉ & Lei Jiang [5]

The pressing crisis of clean water shortage requires membranes to possess effective ion sieving as well as fast water flux. However, effective ion sieving demands reduction of pore size, which inevitably hinders water flux in hydrophilic membranes, posing a major challenge for efficient water/ion separation. Herein, we introduce anomalous water molecular gating based on nanofiltration membranes full of graphene capillaries at 6 Å, which were fabricated from spontaneous π-π restacking of island-on-nanosheet graphitic microstructures. We found that the membrane can provide effective ion sieving by suppressing osmosis-driven ion diffusion to negligible levels (~$10^{-4}$ mol m$^{-2}$ h$^{-1}$); unexpectedly, ultrafast bulk flow of water (45.4 L m$^{-2}$ h$^{-1}$) was still functional with ease, as gated on/off by adjusting hydrostatic pressures within only $10^{-2}$ bar. We attribute this seemingly incompatible observation to graphene nanoconfinement effect, where crystal-like water confined within the capillaries hinders diffusion under osmosis but facilitates high-speed, diffusion-free water transport in the way analogous to Newton's cradle-like Grotthus conduction. This strategy establishes a type of liquid-solid-liquid, phase-changing molecular transport for precise and ultrafast molecular sieving.

The scarcity of drinking water is exacerbated in recent years with more frequent droughts and increasing water pollution caused by climate change and human activities[1–3], urging the innovation of next-generation seawater desalination and industrial water purification technologies for the sustainable supply of fresh water[4,5]. Compared with thermal-based distillation technologies, membrane-based separation technologies are featured with low energy consumption, high efficiency, and are becoming the predominant choice for the advanced water purification approaches[6,7]. However, a formidable obstacle exists and significantly restricts the continuous development of membrane-based technologies, which is how to realize precise and

ultrafast molecular sieving simultaneously[8]. In particular, efficient water-ion separation requires selective sieving of ions while conducting water molecules in a fast mode; nevertheless, under the conventional molecular transport theory, effective ion sieving and ultrafast water flux are two conflict factors. For instance, in hydrophilic nanofiltration (NF) membranes with 2D sub-nano capillaries, such as in graphene oxide membranes (GOMs) and MXene membranes (water contact angles <50°), the effective ion sieving can be achieved by tuning the interspaced channels in the atomic scale, through Coulombic interactions between the oxygenated capillary wall and the inserted ions/molecules[9,10]. However, the rate of water transport will

[1]Henan Institute of Advanced Technology, Zhengzhou University, Zhengzhou 450003, PR China. [2]College of Materials Science and Engineering, Zhengzhou University, Zhengzhou 450001, PR China. [3]School of Chemical Engineering, Zhengzhou University, Zhengzhou 450001, PR China. [4]Department of Chemical and Materials Engineering, University of Alberta, Edmonton, AB T6G 1H9, Canada. [5]Key Laboratory of Bio-inspired Materials and Interfacial Science, Technical Institute of Physics and Chemistry, Chinese Academy of Sciences, Beijing 100190, PR China. [6]These authors contributed equally: Qian Zhang, Bo Gao, Ling Zhang. ✉e-mail: hongbo.zeng@ualberta.ca; qunxu@zzu.edu.cn; xinweic@zzu.edu.cn

also be impeded due to the steric resistance and its hydrogen-bonding interaction with the oxygenated walls[11,12]. Recently, fast frictionless flow of water has been proposed in atomic-scale graphene capillaries (water contact angles ≈ 85°)[13–15], projecting a promising direction for overcoming the obstacle. Unfortunately, in reduced graphene oxide (rGO)-based membranes, the reduced region tends to restack, sealing the atomic-scale graphene capillaries and leaving large graphitic pores for permeation[16]. As such, although water flux is fast, ineffective ion sieving properties have been found on rGO-based membranes[17]. Up to now, it is still exceptionally challenging to control the size of 2D graphene slit pores with angstrom precision in rGO-based membranes. As a compromised strategy, thermal-membrane coupled technologies, such as membrane distillation and pervaporation, have been applied onto GOMs[18], for the transport of vaporized water molecules only through the included atomic-scale, graphene capillaries within GOMs[19]; however, their energy consumption is still a concern. Therefore, a new generation of NF membranes that can make full strength of atomic-scale graphene capillaries under osmosis and at ambient temperature is highly desired, although it has long-suffering trouble of exploration.

Biological ion channels possess capabilities of precise and fast ion sieving, as well as gating ability in responses to external stimuli to sustain biological activities[20–22], rendering us inspirations for the development of NF membranes to address the most challenging issue shown above. Instead of gating ions as reported previously[23–25], the ideal scenario for water-ion separation would be fast and selective gating of water molecules while blocking ion diffusion. However, because of the inevitable diffusion of water and ions under osmosis and small size difference between water molecules (~2.7 Å) and hydrated alkali-metal ions (>6.6 Å)[26], fast gating of water transport without deteriorating ion rejection seems impossible.

Herein, we report an anomalous gating behavior that can gate water molecules selectively and rapidly, and meanwhile, strictly inhibit ion diffusion in forward osmosis. This can be realized because we successfully fabricate metal-organic islands pillared, rGO-based NF membranes with well-defined graphene capillaries mainly at 6 Å. The rGO-based membrane enables us to fully exploit the potential of atomic-scale graphene capillaries under osmosis at room temperature. Theoretical simulations show that the water molecular gating can be attributed to the properties of highly hydrogen-bonded network (2D ordered water) formed under graphene nanoconfinement. The solid nature of the nanoconfined water structure hinders diffusion under osmosis, and in the meantime, facilitates fast and diffusion-free bulk flow of water by analogy with Newton's cradle-like Grotthus conduction. The mechanism triggers simultaneous achievement of ultrafast bulk flow of water at 2.5–45.4 L m$^{-2}$ h$^{-1}$ and negligible ion permeation rates of ~10$^{-4}$ mol m$^{-2}$ h$^{-1}$ for various alkali-metal ions. We suggest that this underlying mechanism could also be generalized in other NF membranes, such as in metal-organic framework (MOF)[27] and covalent-organic framework (COF)-based membranes[28]. This work establishes a type of liquid–solid–liquid, phase-changing molecular transport approach that provides a unique platform for fundamental advancement as well as real-world applications in a broad range of fields.

## Results
### Preparation of NF Membranes full of atomic-scale graphene capillaries
The fabrication of 2D material NF membranes full of atomic-scale graphene capillaries started from the synthesis of a 2D island-on-nanosheet heterostructure. The catalytic reaction of forming amorphous Ni-pPD layered structure requires the presence of oxygen (Supplementary Fig. 1). As the oxygenated functional groups cover about 40–50% area on GO nanosheets[19], we propose to use these functional groups as the only catalytic sites for the synthesis of a 2D

metal-organic material, Ni-p-phenylenediamine (Ni-pPD), resulting in a heterostructure of Ni-pPD monolayer-islands distributed uniformly on rGO nanosheets (Fig. 1a). Energy dispersive X-ray (EDX) mapping images in Fig. 1b and Supplementary Fig 2 show homogeneous distribution of C, Ni, and N on the resulted nanosheets. The functionalities of Ni-pPD were then further investigated by Fourier-transform infrared spectroscopy (FTIR) and X-ray photoelectron spectroscopy (XPS). The characteristic peaks in FTIR (Supplementary Fig. 3) at 1605 cm$^{-1}$, 1486 cm$^{-1}$, 1274 cm$^{-1}$, and 1168 cm$^{-1}$, corresponding to –C=N groups, C=C stretching vibrations of the aromatic benzene rings, –C–N groups, and the existence of quinonoid (Q) rings in N=Q=N groups, respectively. These peaks agree well with those on pure Ni-pPD samples, which suggests the success of growing Ni-pPD on the surface of nanosheets. In addition, the oxygenated functional groups have been substantially removed in the resulted nanosheets, implying the reduction of GO to rGO after the reaction. XPS analyses (Supplementary Figs 4 and 5) also show the characteristic peaks of Ni-pPD (–C–N and –C=N groups in C 1s, –NH– and –N= groups in N 1s)[29] as well as the reduction of GO in C 1s, consistent with the observations found in FTIR results. Moreover, zeta potential (Supplementary Fig. 6) has been substantially reduced from −33 mV for GO to −10 mV for the composite nanosheets, further proving the reduction of GO. Therefore, the above results demonstrate that the synthesized materials are Ni-pPD@rGO nanosheets. Since Ni is the only heavy metallic element in Ni-pPD@rGO nanosheets, the bright area in high-angle annular dark-field (HAADF) image (Fig. 1c) represents the distribution of Ni-pPD, which, as expected, demonstrates a pattern of 2D nano-islands on the surface of rGO. High-resolution transmission electron microscopic (HRTEM) image (Fig. 1d) also illustrates the configuration of 2D, amorphous Ni-pPD nano-islands on top of rGO, which agrees well with HAADF result. We have also performed density functional theory (DFT) to simulate the microstructure of Ni-pPD@rGO nanosheets. The top-view and cross-view of the optimized microstructure can be seen from Fig. 1e. As compared with the heterostructure before relaxation (Supplementary Fig. 7), Fig. 1e demonstrates that the four benzene rings around Ni ions in Ni-pPD tend to be planarized by rGO through π–π stacking.

It is known that, upon reduction of GO, the restacking should occur instantaneously, leaving no free space between the restacked rGO layers[30]. This is one of the key reasons that prevent us from precisely tuning the nanosize of graphene capillaries in pure rGO membranes. Interestingly, because of the 2D island-on-nanosheet graphitic microstructure obtained in this work, the unavoidable restacking of the Ni-pPD@rGO nanosheets is an asset in creating atomic-scale graphene capillaries. Atomic force microscopic (AFM) image (Fig. 2a) shows that the thickness of the restacked Ni-pPD@rGO nanosheets is around 2.88 nm. Considering a monolayer of Ni-pPD nano-islands formed on each side of rGO, the repeating unit of the restacked structure between two rGO should be around 1 nm. The AFM analysis then suggests that there are 3 repeating units in the restacked Ni-pPD@rGO nanosheets after the synthesis, with the thickness of each unit being around 0.96 nm. This value has been confirmed by HRTEM images in Fig. 2b, c. From the edge of the restacked Ni-pPD@rGO nanosheets, it is clear to see that the interlayer spacing between two neighboring rGO is 0.96 nm. More importantly, as also indicated in Fig. 2c, atomic-scale graphene capillaries have been created because of the restacking of these 2D island-on-nanosheet heterostructures. In order to get a clear image of the pore structure, we have conducted DFT simulations for the repeating unit of the restacked Ni-pPD@rGO nanosheets. The optimized structure computed by DFT (Supplementary Fig. 8) demonstrates that two layers of Ni-pPD are pillared between two rGO through π–π stacking with all the stacking distances being in the range from 0.34 to 0.36 nm, generating the interlayer spacing of 0.96 nm from the top to the bottom. As such, the d-spacing between the 1st-layer rGO and 3rd-layer pPD was calculated to be 6.0 Å,

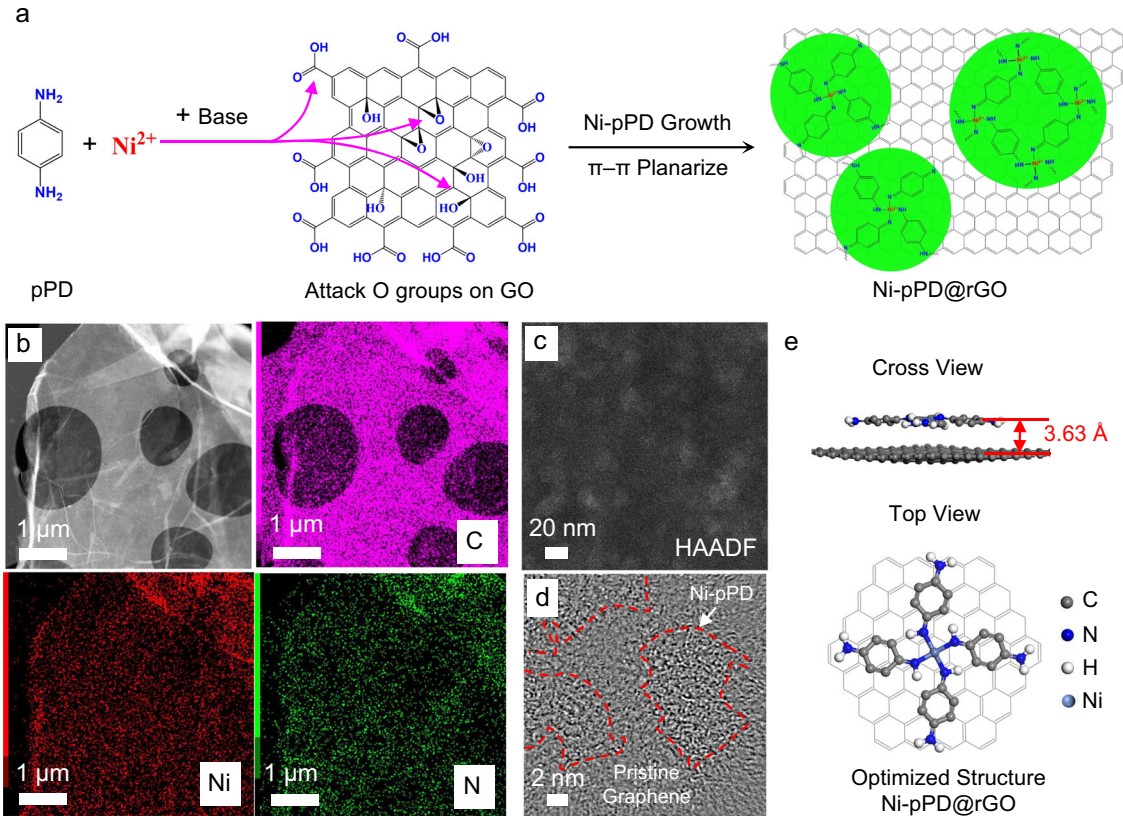

**Fig. 1 | Synthesis and characterization of Ni-pPD@rGO nanosheets. a** Synthesis route of Ni-pPD@rGO nanosheets using the oxygenated functional groups as the catalytic sites for the reaction. **b** STEM-EDX mapping shows the uniform distribution of C, Ni, and N on Ni-pPD@rGO nanosheets. HAADF (**c**) and HRTEM (**d**) images of Ni-pPD@rGO nanosheets illustrating the distribution of 2D Ni-pPD nano-islands on rGO nanosheet microstructure. **e** DFT simulation shows the optimized structure of Ni-pPD@rGO nanosheets, demonstrating that Ni-pPD tends to be planarized by rGO through π−π stacking.

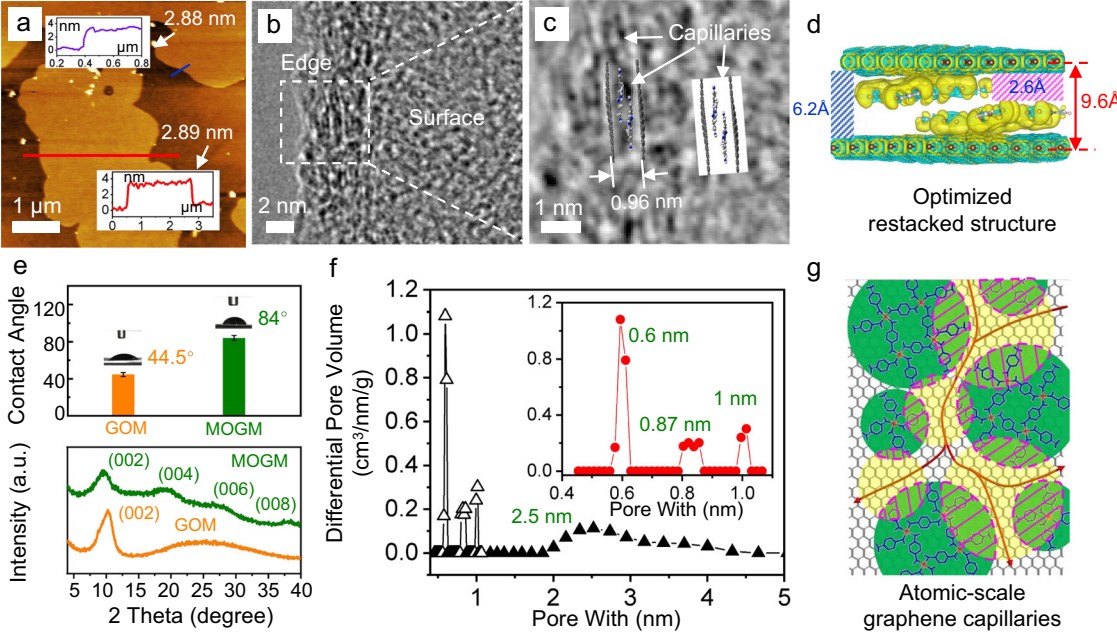

**Fig. 2 | Characterization of restacked Ni-pPD@rGO nanosheets and MOGMs. a** AFM image shows the spontaneously restacked Ni-pPD@rGO nanosheets. **b, c** HRTEM images of the edge of restacked Ni-pPD@rGO nanosheets, illustrating the interlayer spacing of 0.96 nm between two neighboring rGO and atomic-scale graphene capillaries created by restacking. **d** The optimized structure of restacked Ni-pPD@rGO nanosheets computed by DFT, with the interlayer spacing labeled by a red arrow and the free space marked by slash lines. **e** Water contact angle and XRD spectra of MOGMs and GOMs. **f** Pore size distribution of MOGMs. **g** Schematic of the sub-nano pathways to conduct water molecules in MOGMs. Green: bottom Ni-pPD nano-islands; Yellow: top Ni-pPD nano-islands; Pink-shaded frames: fully blocked regions by the restacking of two Ni-pPD nano-islands.

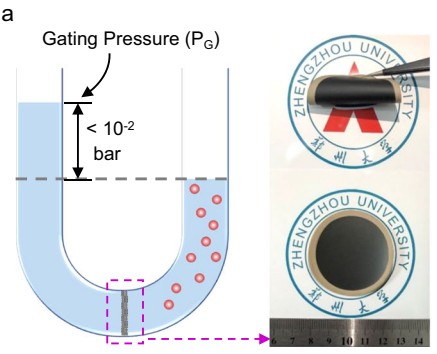

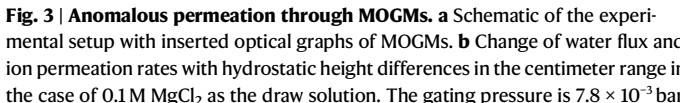

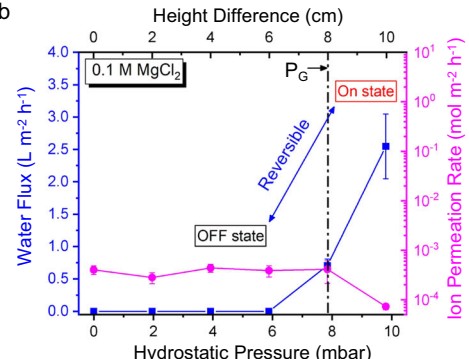

**Fig. 3 | Anomalous permeation through MOGMs. a** Schematic of the experimental setup with inserted optical graphs of MOGMs. **b** Change of water flux and ion permeation rates with hydrostatic height differences in the centimeter range in the case of 0.1 M MgCl$_2$ as the draw solution. The gating pressure is $7.8 \times 10^{-3}$ bar

(labeled as a dashed line). The reversible water on/off switch, ultrafast water flux, and effective ion sieving for Mg$^{2+}$ ions are also illustrated. All the error bars in this figure represent the standard deviations for at least three measurements.

giving the free space of 2.6 Å (marked as pink slash lines in Fig. 2d); and also, the *d*-spacing between the 1st-layer rGO and 4th-layer rGO was determined to be 9.6 Å, giving the free space of 6.2 Å (marked as blue slash lines in Fig. 2d). Therefore, although the restacked rGOs should not present free space, the rGOs pillared by Ni-pDP nano-islands can sufficiently maintain their distance for molecular transport.

The membranes made of the restacked Ni-pPD@rGO nanosheets have then been fabricated by vacuum filtration, followed by the controlled evaporation under external pressures. The detailed process has been illustrated in Supplementary Fig. 9. The densely packed Ni-pPD@rGO nanosheets have been demonstrated in these micrometer-thick membranes, with a uniform distribution of the contained elements obtained by scanning electron microscopic (SEM) and EDX mapping images taken from the top (Supplementary Fig. 10) and the cross section (Supplementary Fig. 11). These membranes are, therefore, denoted as metal-organic (Ni-pPD) pillared, graphene-based membranes (MOGMs). As consistent with previous reports, GOMs contain hydrophilic capillaries and the contact angle obtained for GOMs is 44.5° (Fig. 2e). In contrast, the contact angle for MOGMs is 84°, which is close to the contact angle of water on graphite (85°)[31], indicating that MOGMs are full of atomic-scale graphene capillaries. In addition, GOMs normally present only one X-ray diffraction (XRD) peak at around 10.4° (~0.85 nm) as also shown in Fig. 2e, and pure rGO membranes normally present only one XRD peak at the interspacing around 25° (~0.35 nm) because of π-π stacking[30]. The XRD result obtained for MOGMs, however, demonstrates four distinct peaks of (002), (004), (006), and (008), with the interspacing of (002) being 0.93 nm and that of (006) being 0.32 nm, strongly supporting our conclusions about the restacked repeating unit structure of Ni-pPD@rGO nanosheets. Importantly, as predicted in Fig. 2d, graphene capillaries within MOGMs should have specific sub-nano sizes. This has been proved by Brunauer-Emmett-Teller (BET) results in Fig. 2f and Supplementary Fig. 12. The pore size distribution (PSD) analysis was conducted using the slit pore geometry and by applying a hybrid nonlocal DFT (NLDFT) kernel, where N$_2$ gas was used for the micropores larger than 1 nm and CO$_2$ gas was used for the ultra-micropores smaller than 1 nm (the lower limit is 0.35 nm). Figure 2f clearly demonstrates three sharp and distinctive peaks below 1 nm, with the strongest peak sitting at ~0.6 nm. Considering the calculated free space of 6.2 Å in Fig. 2d, the PSD experimental results in Fig. 2f correspond well with our DFT calculations. Unlike activated graphene/carbons that normally have a continuous size distribution of micropores[32], the pore structure in MOGMs resembles the structural characteristics of molecular sieves with the pathways to conduct water molecules illustrated in Fig. 2g, manifesting the delicacy of this synthesis strategy.

## Anomalous permeation under osmosis

Conventional understanding has it that, in a U-shape permeation system (Fig. 3a) where a concentration gradient is present, water from feed side will diffuse into the salt solution on the draw side through a semipermeable membrane, driven by osmotic pressure[33]. This trend has been clearly presented in Supplementary Fig. 13, by using 750-nm-thick GOMs in the permeation test and the salts of 0.1 M of KCl, NaCl, and MgCl$_2$ on the draw side. In addition, since the osmotic pressures are at the level of 5-8 bar, small hydrostatic pressures applied at the level of ~10$^{-2}$ bar onto the feed side (also illustrated in Fig. 3a) will not affect the ion permeation performance appreciably for GOMs (Supplementary Fig. 13).

We examined an unexpected permeation behavior on MOGMs (Fig. 3b). The optical graphs of MOGMs can also be seen in Fig. 3a. Under the osmotic pressure of 7.4 bar (0.1 M MgCl$_2$ solution) and with no height difference between two compartments in the U-shaped cell ("OFF" state in Fig. 3b), no water flux can be detected for more than 1 day of operation, indicating that the diffusion of water has been substantially suppressed. This also confirms the integrity of MOGMs during their fabrication. However, when exerting a small hydrostatic pressure as low as $7.8 \times 10^{-3}$ bar on the feed side (indicated as a dashed line in Fig. 3b), the water flow suddenly shoots up to 0.71 L m$^{-2}$ h$^{-1}$. The water flux keeps increasing up to 2.5 L m$^{-2}$ h$^{-1}$ at $9.8 \times 10^{-3}$ bar ("ON" state in Fig. 3b), which is about 10 times faster than that obtained by GOMs under the same experimental conditions (Supplementary Fig. 13). The experimentally detected rates of water flow through the atomic-scale graphene capillaries in MOGMs are more than two orders of magnitude higher than the rate of water flow under the diffusion mode, estimated by considering a typical self-diffusion coefficient of neat water (≈10$^{-10}$ m$^2$ s$^{-1}$)[34] (Fig. 3b). Thus, we suspect the diffusion-free, bulk flow of water occurred in the "ON" state, which will be illustrated later. Because of the blockage of water diffusion, ion diffusion has also been suppressed significantly. The ion permeation rates of Mg$^{2+}$ show very low values (Fig. 3b), around 3 orders of magnitude lower than those of GOMs (Supplementary Fig. 13). In particular, at the hydrostatic height difference of 10 cm, the Mg$^{2+}$ ion permeation rate is lower than the detection limit of inductively coupled plasma mass spectrometry (ICP-MS). Similar phenomena can also be found for other hydrated alkali-metal ions (Supplementary Figs 14 and 15).

In Fig. 3 we see that water hardly permeates by diffusion under a large osmotic pressure of 7.4 bar, but fast permeates under a small hydrostatic pressure at the level of 10$^{-2}$ bar. More importantly, water flow can be turned on and off reversibly (Fig. 3b), by adjusting the hydrostatic pressure higher and lower than the gating pressures (P$_G$: ~10$^{-2}$ bar), respectively, revealing the successful establishment of a

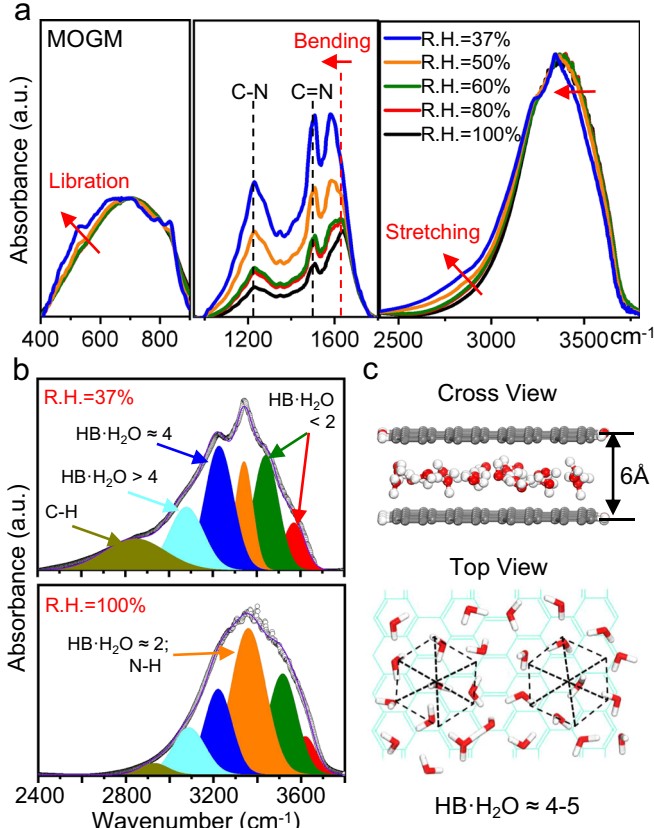

**Fig. 4 | Arrangement of water molecules within atomic-scale graphene capillaries of MOGMs. a** In-situ FTIR for the absorbance of water inside MOGMs from 37% to 100% of environmental relative humidity (R.H.). Water content increases from top to bottom. Different spectral ranges are shown to highlight vibrational modes of intercalated or adsorbed water. **b** Gaussian fits of five OH stretching modes shown in cyan, blue, orange, green, and red from lower frequencies to high frequencies, illustrating the decreasing number of H-bonds for $H_2O$ molecule (HB·$H_2O$) in average. **c** DFT calculations for the optimized arrangement of water molecules within the graphene capillaries with the interspacing of 6.0 Å, showing a close to hexagonal-type ordered structure with the HB·$H_2O$ being about 4 and 5. Red: oxygen atoms, white: hydrogen atoms, gray or cyan grids: graphene layers.

gating mechanism for precise and ultrafast molecular sieving. It is also important to note that ion diffusion under osmosis for MOGMs has been highly suppressed, but not completely stopped. Especially in the "OFF" state, ion permeation rates decrease appreciably with the increasing size of hydrated ions (Supplementary Fig. 16). It implies that size exclusion effect plays a key role and partial dehydration of ions should occur in this case[35], which results in high activation energy barriers for ion permeation through the atomic-scale graphene capillaries. The highly suppressed ion permeation rates also exclude the possibility of in-situ generation of defects or cracks in the membranes after applying the small gating pressures onto the feed side.

### Characterization of ordered water structure within atomic-scale 2D graphene capillaries

To explicate the underlying physical mechanism observed in Fig. 3, the H-bonded water network confined within the atomic-scale graphene capillaries have been investigated first. We performed in-situ FTIR on MOGMs (Fig. 4) and GOMs (Supplementary Fig. 17). Figure 4a displays three absorption bands of $H_2O$ from MOGMs emersed in the environment with changing relative humidity from 37% to 100%, which are the OH stretching band between 3000 and 3700 cm⁻¹, the water bending mode around 1625 cm⁻¹, and its libration mode between 400

and 900 cm⁻¹[36]. Membranes in 100% humidity (immersed in pure water over 1 h) mainly represent the structure of bulk liquid water, while those in 37% humidity (environmental humidity) more represent the water structure within the atomic-scale graphene capillaries. Figure 4a shows that all three $H_2O$-related absorption bands shift to lower frequencies with decreasing humidity, while the peaks not related to $H_2O$ (C−N and C=N) maintain their peak positions, indicating that H-bonded water network is stronger within the atomic-scale graphene capillaries than that in the bulk liquid water[37].

Gaussian fitting of the spectra in Fig. 4b shows that the OH stretching band can be resolved into five distinct components, corresponding to different number of hydrogen bonds (HB) associated averagely with each $H_2O$ molecule[38]. The orange peak sitting at around 3400 cm⁻¹ and the blue peak at around 3220 cm⁻¹ are related to 2-coordinated H-bonded water (HB·$H_2O$ ≈ 2) and 4-coordinated H-bonded water (HB·$H_2O$ ≈ 4), respectively[39]. For instance, the peak of water in ice sits around 3250 cm⁻¹ with HB·$H_2O$ ≈ 4, and that of bulk liquid water sits between 3200 and 3400 cm⁻¹ with HB·$H_2O$ ≈ 3[36]. In addition, the green and red peaks correspond to loosely bonded water with HB·$H_2O$ < 2, determined to be the characteristic peaks of one-dimensional water chain within single-walled carbon nanotubes and metal-organic frameworks[36,38]. Therefore, the cyan peak in Fig. 4b corresponds to strongly bonded water with HB·$H_2O$ > 4. Particularly in Fig. 4b, the blue and cyan peaks are predominant for the MOGMs in 37% humidity, also suggesting that H-bonded water network within the atomic-scale graphene capillaries is stronger than those in both bulk liquid water and ice.

DFT computations have then been conducted to simulate the 2D H-bonded water network within the graphene capillaries with two different interspacings, one with 6.0 Å (Fig. 4c) and the other one with 8.7 Å (Supplementary Fig. 18). After relaxation, the optimized arrangements of water molecules within 6.0 Å-interspaced graphene capillary (Fig. 4c) show a monolayer of 2D ordered structure. Considering small vibrations of water molecules in the ordered structure, here, we found that this structure has an approximately hexagonal type of unit cell, with the HB·$H_2O$ being about 4 and 5 (Supplementary Fig. 19), illustrating the blue and cyan peaks presented in-situ FTIR (Fig. 4b). In addition, the density of water within the confined space has been determined to be 2.2 g cm⁻³, which is consistent with the density of water in GO membranes measured before[40]. Furthermore, DFT calculations also show that a double-layer, hexagonally arranged structure exists in 8.7 Å-interspaced graphene capillary (Supplementary Fig. 18) with the density of water also determined to be 2.2 g cm⁻³.

The exposed graphene surface has the wetting property with a contact angle of 84°, which can be explained by the relatively weak van der Waals attraction between water and graphene. However, within the atomic-scale graphene capillaries, this short-range attraction plays a key role and a large capillary pressure can be generated, which has been estimated to be in the order of 1000 bar[13]. Hence, it should be the large capillary pressure that compacts the distance between water molecules and induces the formation of close to hexagonally arranged water nanosheet in Fig. 4. Note that edge carbon atoms and dangling bonds are also present in the micrometer-thick MOGMs, which may add resistance to the molecular transport. However, the in-situ FTIR results in Fig. 4a, b prove that those external defective microstructures did not hinder the formation of the ordered water nanosheet within the atomic-scale graphene capillaries in MOGMs.

### Underlying mechanism of liquid-gating permeation

Based on the highly connected, H-bonded water network within the confined graphene space, we have then constructed the permeation cell under osmosis and conducted MD simulations (see "Methods" in supporting materials). The "OFF" state is illustrated in Fig. 5a and Supplementary Movie 1. As indicated by the dashed lines in Fig. 5a, the water molecules sitting along these lines act as the wall, so that the

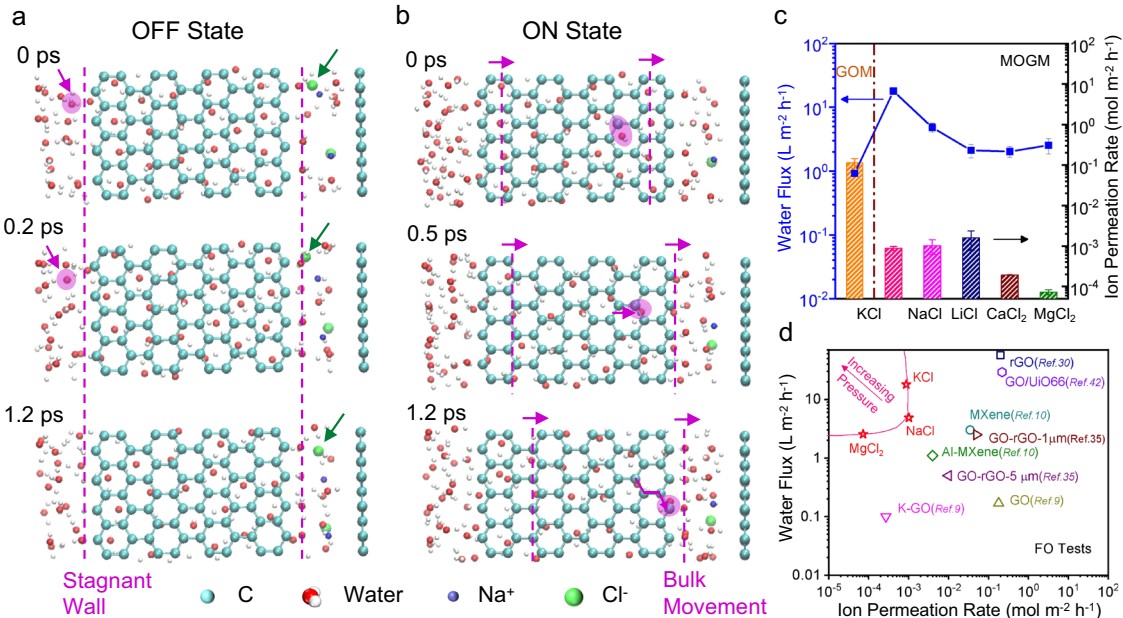

**Fig. 5 | Theoretical computations for the underlying mechanism of liquid-gating permeation and its application in forward osmosis (FO). a** Three sequential snapshots of MD simulation in the "OFF" state illustrating the blockage of any attempts of water (pink arrows) and ion (green arrows) diffusion. **b** Three sequential snapshots of MD simulation in the "ON" state illustrating the bulk movement of water molecules (pink arrows). Ultrafast water flux and effective ion sieving achieved simultaneously for various ions on MOGMs (**c**), in comparison with those of the reported membranes in the literature for forward osmosis (**d**). All the error bars in this figure represent the standard deviations for at least three measurements.

attempts of water or ion diffusion (labeled by arrows) will be bounced back. Since the formation of crystal-like water nanosheets, the whole water-intercalated graphene capillary becomes a solid, and the dashed lines are the solid water/liquid water interface. Thus, the water molecules sitting at the interface should still be tightly bonded by the inner ordered water structure, forming walls that suppress the diffusion of water molecules. It is then further suggested that both this solid water/liquid water interface and partial dehydration energy barrier result in the significant suppression of ion diffusion.

We continue the simulation by adding more water molecules in the feed side (left) to simulate "ON" state (Fig. 5b and Supplementary Movie 2). In this case, water molecules transport fast through the graphene capillaries to the draw side, in the way that the ordered water nanosheet slides to the right as whole (bulk flow), as indicated by the dashed lines with changing positions in Fig. 5b. It also excludes the possibility of diffusion because no random motion of water molecules is present in the "ON" state. In addition, we calculated the energy barriers for water transport using DFT (Supplementary Fig. 20). Water transmission along C−C bonds on a graphene sheet gives the lowest energy barrier, which is only 3 meV. In comparison, the energy barrier for the transport of lithium ions on the surface of graphene is 260 meV[41]. The preference of water transmission along C−C bonds has also been highlighted in Fig. 5b and Supplementary Fig. 21a, which can also be seen in Supplementary Movies 2 and 3. It is, therefore, a clear implication that a small exerted pressure should be sufficient to open the gate for fast and diffusion-free bulk water transport.

Taking advantage of this liquid-solid-liquid, phase-changing molecular transport, ultrafast water flux and effective ion sieving can be achieved simultaneously for various ions including $K^+$, $Na^+$, $Li^+$, $Ca^{2+}$, and $Mg^{2+}$, in forward osmosis (Fig. 5c), with the hydrostatic pressures applied at 4, 6, 8, 10, and 10 cm, respectively (Supplementary Fig. 22). This performance is already the best among all the results previously reported on forward osmosis (Fig. 5d and Supplementary Table 1)[9,10,30,35,42], not to mention the case for further increased driving pressures (the arrow in Fig. 5d). For instance, Supplementary Figs 14 and 15 show that water flux can be further improved to 45.4 m⁻² h⁻¹ and

7.1 L m⁻² h⁻¹ with the ion permeation rates of ~10⁻⁴ mol m⁻² h⁻¹ at the hydrostatic pressure of $9.8 \times 10^{-3}$ bar (the height difference of 10 cm) for 0.1 M KCl and NaCl solutions, respectively. The superior anti-swelling properties of MOGMs have also been demonstrated in Supplementary Fig. 23.

## Discussion

Based on all the discussion above, the whole system acts as special communicating vessels, where random motion (diffusion) of water molecules or hydrated ions across the capillary will be suppressed due to the large capillary pressure and the 2D ordered water nanosheets formed within (Fig. 6a). However, no matter how high the internal capillary pressure is, a small hydrostatic pressure can generate ultrafast water flux to balance out the level in two compartments. When applying a hydrostatic pressure, a shear force will be applied onto the ordered water nanosheet. In Supplementary Movie 4, it is clear to see pressure transmission from the high-pressure side to the low-pressure side. Hence, the water transport mechanism in the "ON" state is by analogy with Newton's cradle-like Grotthus conduction (Fig. 6b). Following this way, at any instant time, there is only one line of water molecules that break their original bonding with graphene, which minimizes the energy requirement for the sliding of the ordered water nanosheet. The bulk movement of water molecules in the "ON" state is then the outcome of the continuous mode of Newton's cradle-like Grotthus Conduction. Therefore, although the micrometer-thick MOGMs have long and torturous permeation paths, it is this high-speed, Grotthus conduction that enables ultrafast water flux. Obviously, in the "ON" state, diffusion under osmosis is still highly suppressed.

Moreover, the resistance to this leveling effect should result in the gating pressures. Since the energy barrier is low for the sliding of the ordered water nanosheets (Supplementary Fig. 20), this resistance should be originated from external microstructural reasons. In addition to the edge carbon atoms, dangling bonds, and the interstitial water/ions outside of the graphene capillaries that could add resistance to the bulk water flow, Supplementary Fig. 22 also implies that the counter motion of ions, although suppressed, also exert resistance to the sliding

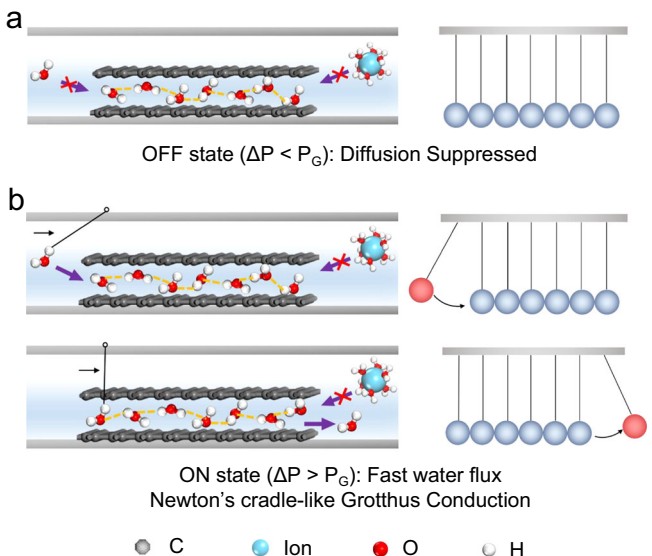

**Fig. 6 | Microscopic illustrations of water molecular gating mechanism. a** "OFF" state. **b** "ON" state with Newton's cradle-like Grotthus Conduction originated from highly connected hydrogen-bonded network.

a

OFF state (ΔP < P$_G$): Diffusion Suppressed

b

ON state (ΔP > P$_G$): Fast water flux
Newton's cradle-like Grotthus Conduction

○ C    ● Ion    ● O    ○ H

of the ordered water nanosheets. All these factors can contribute to the magnitude of gating pressures (Supplementary Fig. 24). In this way, a water molecular gating mechanism has been established for the anomalous permeation behavior being observed in Fig. 3.

This work enriches our fundamental understandings in the correlation between the pore structure and the permeation performance at the atomic scale. In a broader context, the discovered water molecular gating mechanism originated from liquid-solid-liquid, phase-changing molecular transport not only suggests ways of prohibiting the unavoidable diffusion in the liquid state, but also provides solutions for enhancing the sluggish transport kinetics in the solid state, which will be of great benefits for the applications in a variety of fields related to molecular transport, ranging from water desalination and purification to life science, nanofluidics, ion transistors, energy conversion, energy storage and beyond.

## Methods

### Materials
Nickel nitrate hexahydrate (Ni(NO$_3$)$_2$·6H$_2$O) and p-phenylenediamine (pPD) were purchased from Sigma Aldrich, concentrated ammonium hydroxide solutions (25-28 wt.%) (NH$_4$OH) was purchased from Shanghai Macklin Biochemical Co. Dopamine hydrochloride, trizma® base, potassium chloride (KCl), sodium chloride (NaCl), lithium chloride (LiCl), calcium chloride (CaCl$_2$), magnesium chloride hexahydrate (MgCl$_2$·6H$_2$O) were purchased from Aladdin. Polyethersulfone (PES) substrates (pore size of 0.22 μm) were purchased from Merck Millipore Ltd. Milli-Q water (18 mΩ) was made by PUXI GWB-2. All of the analytical reagents mentioned above were used without further purification.

### Synthesis of restacked Ni-pPD@rGO nanosheets
GO suspensions (5 mg/ml) were first fabricated according to the modified Hummer's method[43,44]. A solution of 1.92 g (6.6 mmol) of Ni(NO$_3$)$_2$·6H$_2$O in 28 ml of Milli-Q water was mixed with 32 ml GO suspension in a sealed three-necked flask. Another solution of 1.57 g (14.5 mmol) of p-phenylenediamine (pPD) (Sigma Aldrich) in 60 ml of Milli-Q water was also prepared in a sealed flask. Two solutions were separately stirred and purged with Ar gas for 1 h to exclude any external oxygen sources in the solutions. Two solutions were then mixed together in the three-necked flask and 20 ml of concentrated

NH$_4$OH solution was added drop by drop to initiate the reaction. During the reaction, the mixture was stirred in a sealed flask with Ar gas bubbling for over 10 h at room temperature. The resulted black powders were filtered and then washed with Milli-Q water and ethanol thoroughly. Finally, the solids were redispersed into 200 ml Milli-Q water to make Ni-pPD@rGO suspensions (0.6 mg ml$^{-1}$). For comparison, Ni(OH)$_2$@GO nanosheets were also synthesized by treating GO only with Ni(NO$_3$)$_2$ and NH$_4$OH (without adding the pPD solution).

### Preparation of Ni-pPD@rGO membranes (MOGMs) and GOMs
First, PES substrates were coated with polydopamine (PDA) to strengthen the adhesion between the membranes and the substrates, and to make sure the integrity of the membranes during the permeation tests, according to previous report[5]. The coating solution (2 mg mL$^{-1}$) was prepared by dissolving 0.16 g of dopamine hydrochloride into 80 mL of 0.01 M Tris buffer at pH = 8.5. The PES substrates were then immersed in this solution at room temperature for 2 h. After that, the substrates were rinsed in DI water and dried in an oven at 60 °C. 5 ml Ni-pPD@rGO suspension was diluted into 50 ml, sonicated for 5 min, and then vacuum filtrated under 0.98 bar to form MOGM on a PDA-coated PES substrate. For comparison, 0.2 ml GO suspension (5 mg ml$^{-1}$) was also diluted into 50 ml, sonicated for 5 s, and then vacuum filtrated under 0.98 bar to form GOM on a PDA-coated PES substrate. After that, both MOGM or GOM on PDA-PES membranes were immediately clamped between two glass plates and dried under vacuum with controlled temperatures. The controlled evaporation is an essential step to make sure MOGMs with well-defined, atomic-scale graphene capillaries at 6 Å. A freestanding MOGM was fabricated using the similar procedure but vacuum filtrated directly on bare PES substrates without PES coating, from which MOGM can be easily peeled off after drying. The freestanding MOMGs were used for BET and PSD analyses.

### Characterizations
Scanning electron microscopy (SEM) imaging were performed on JSM-7500F Field Emission SEM operated at high vacuum with a resolution of ~1.0 nm and Hitachi SU8100 equipped with Elemental X-ray spectroscopy (EDX) from EDAX Inc. Transmission electron microscopy (TEM) images were acquired on JEM-2100 at 200 kV, where EDX mapping was conducted under scanning TEM (STEM) mode. High-angle annular dark-field (HAADF) image was acquired on FEI Titan krios at 300 kV. Atomic force microscopy (AFM) imaging was performed on the SHI-MADZUSPM9700 AFM microscope. The XRD patterns were detected by Y-2000 X-ray Diffractometer with copper Kαradiation (λ = 1.54 Å), using the powder mode. XPS spectra was achieved on ESCLAB 280 system with Al/K (photon energy = 1486.6 eV) anode mono X-ray source. Fourier-transform infrared spectroscopy (FTIR) was performed by Bruker VERTEX 70 spectrometer. Contact angle was performed on Biolin Scientific Attension Theta Flex. Zeta potential was performed on Malvern zeta sizer nano series, with Ni-pPD@rGO nanosheets, GO nanosheets, and pure Ni-pPD particles being dispersed in Milli-Q water in the concentration of 0.05 mg ml$^{-1}$. Gas-adsorption measurements were made using Micromeritics ASAP 2020 using N$_2$ gas at 77 K as well as carbon dioxide at 273.2 K. Before the gas-adsorption experiment, freestanding MOGMs were thoroughly washed and dried, and then "outgassed" under vacuum at 150 °C for 8 h.

### Permeation performance measurements
The water and ion permeation through the membranes were measured on a home-made U-shape device with an effective area of 7.07 cm$^2$ at room temperature. Take the permeation tests of Mg$^{2+}$ as an example: 100 ml Milli-Q water and 100 ml 0.1 M MgCl$_2$ aqueous solution were added into the feed side and draw side, respectively. During the test, the MOGM layers were facing to the feed side, using the compressive pressure to make sure the integrity of the membrane during the

permeation test, although the opposite facing way was also tested and gave us the same liquid-gating phenomenon. Magnetic stirrings were applied to both sides of the feed and the draw solutions. Inductive coupled plasma emission spectrometer (ICP) (Perkin Elmer Avio 500) was used obtain the cation concentrations in solutions. Each permeation test was conducted for over 4 h and repeated for at least three times to obtain reliable data. The test temperature was about 25 °C.

The water flux $J$ (L m$^{-2}$ h$^{-1}$) and the ion permeation rate $R$ (mol m$^{-2}$ h$^{-1}$) can be calculated by according to Eqs. (1) and (2), respectively:

$$J = \frac{\Delta V_d}{A \Delta t} \tag{1}$$

$$R = \frac{C_f V_f}{A \Delta t} \tag{2}$$

where $\Delta V_d$ is the volume change of draw solution, $C_f$ is the concentration of ions detected by ICP in the feed side, $V_f$ is the volume of water in the feed side, $A$ is the effective membrane area, and $\Delta t$ is the test time.

### In-situ FTIR measurements for MOGMs and GOMs
In situ FTIR measurements were performed using Bruker's INVENIO R spectrometer with FTIP range: 400 – 4000 cm$^{-1}$ at 4 cm$^{-1}$ resolution. The central hole of the sample holder was covered by freestanding membrane samples (no substrate), which were then placed at normal incidence relative to the incident beam inside of the test chamber at 298 K. A dehumidifier was applied to maintain the environment at relative humidity (RH) of 37% for the initial tests. Then the membranes were immerged in Milli-Q water for over 1 h and tested with full of water as RH of 100%. During water evaporation, the membranes were tested at different RH values of 80%, 60%, and 50%. A humidity meter was used inside the test chamber to check the values of RH.

### XRD measurements for anti-swelling properties
The MOGMs and GOMs on the PDA-coated PES substrates were immersed in KCl, NaCl, LiCl, CaCl$_2$, and MgCl$_2$ solution of 0.1 mol L$^{-1}$ (M) concentration at room temperature for 14 days. Then the wet membranes with salt solution were taken out and quickly characterized by XRD from 3° to 15°. Same tests were run for dried MOGMs and GOMs as control samples.

### Geometric structure optimization
The spin-polarized density functional theory (DFT) calculations were performed using the Vienna Ab initio Simulation Package (VASP) code[45–47]. The generalized gradient approximation (GGA) of the Perdew–Burke–Ernzerhof (PBE) functional with van der Waals correction was applied to optimize the geometric structures[48]. The interactions between the ions and valence electrons were described by Projector augmented wave (PAW) potentials[49,50]. The force on each atom was less than 0.01 eV/Å, and the convergence criteria of the total energy for all the calculations were set as $1 \times 10^{-5}$ eV. A plane wave cutoff energy of 500 eV was chosen for all calculations.

### Ab initio molecular dynamics (MD) simulations
The simulation system includes periodic graphene sheets, a particular amount of water molecules, and NaCl ions in a periodic box with the dimensions 29 × 11 × 9 Å or 29 × 11 × 11 Å. The Nosé thermostat was used to run all the simulations in the canonical ensemble (NVT), which has a constant number of atoms, volume, and temperature[51,52]. Newton's equations of motion were integrated using Verlet's logarithm in the velocity form with a time step of 1 fs for all the structure. In the simulation, the period boundary conditions were applied to all three dimensions. Graphene nanosheets were fixed in order to prevent them from interfering with the movement of water molecules and NaCl ions.

Graphene nanosheets put horizontally create a sub-nano-confined capillary, while graphene nanosheets positioned vertically divide the space into two parts of equal volume and create two independent cubic boxes, forming a permeation cell. To achieve varied system pressures and solution concentrations, different numbers of water molecules and NaCl ions were deposited in two identical volume areas separated by vertically positioned graphene nanosheets. For data analysis and molecular visualizations, the application VMD[53] was utilized.

## Data availability
All data supporting this study and its findings are available within the article and Supplementary Information. Additional supporting data of this study are available from the corresponding authors on request. figshare. Dataset. https://doi.org/10.6084/m9.figshare.22679818. Source data are provided with this paper.

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

## Acknowledgements

The authors gratefully acknowledge the financial support from National Natural Science Foundation of China (Nos. 52272242, 21773216 and 51173170) and Distinguished Young Researchers' Program of Zhengzhou University (No. 32310221). H.Z. acknowledges the support from the Natural Sciences and Engineering Research Council and the Canada Research Chairs Program. We also would like to acknowledge the National Supercomputing Center in Zhengzhou for the assistance of theoretical computations.

## Author contributions

H.Z., L.J., X.C., Q.X., conceived the project and guided the research; X.C., Q.Z., L.Z, L.X., designed and conducted the permeation and ICP experiments; B.G performed the computational modeling; J.C. and Y. C. assisted with building the experimental setup and sample characterization; X.C, Q.X., H.Z., L.J. wrote and revised the paper with inputs from all authors.

## Competing interests

The authors declare no competing interests.
