## [Peer Review File · Nature Communications]

Anomalous water molecular gating from atomic-scale graphene capillaries for precise and ultrafast molecular sievingREVIEWER COMMENTS

Reviewer #1 (Remarks to the Author):

The manuscript entitled "Anomalous Liquid Gating from Atomic-scale Graphene Capillaries for Precise and Ultrafast Molecular Sieving" has many issues which need to be rectified before submitting it to any scientific journal. At the current stage, it is not suitable for publication in Nature Communication. Some (not all) of the major issues are listed below.

1. The claim "for the first time, allows us to take full advantage of atomic-scale, graphene capillaries under osmosis at room temperature" is misleading. Firstly, the manuscript is on reduced graphene oxide capillaries. Secondly, one of the authors Prof Lei Jiang group are already pioneer in the power generation from GO membranes, including osmosis. Hence the claim is inappropriate.

2. It is quite surprising to know that the oxygenated functional groups disappear in the resulting nanosheets. GO nanosheets are known to have different oxygen-containing functional groups like -O-, -OH, -COOH, C=O, etc. It is very hard to reduce all these functional groups especially the -O- and -COOH even in harsh conditions. In the FTIR compared in the supplementary Figure 3, the authors have indicated the presence of the OH and the C-OH groups in the Ni-pPD@rGO nanosheets, which directly contradicts their claim of "the oxygenated functional groups disappear in the resulted nanosheets, implying the reduction of GO to rGO after the reaction." Careful observation in the spectra also indicate the presence of other oxygen functional groups like C-O-C in the Ni-pPD@rGO nanosheets. In addition, the XPS (supplementary Figure 4a) spectra of the Ni-pPD@rGO nanosheets have a very strong peak for O 1s at ~530 eV. This clearly indicates the presence of different oxygen functional groups.

3. How did the authors confirm the formation of the Ni-pPD complex? They did not discuss it properly in any of the characterization techniques. For example, in the FTIR, which peak confirms the Ni-N bond (N=Q=N peak?). In XPS, the peak under O 1s at ~530 eV is not well described. In all the characterization techniques, the authors compared the spectra of the GO and Ni-pPD@rGO nanosheets. Although the formation of the Ni-pPD complex is a known reaction (as mentioned in the ref 26), but this formation might be different in the presence of the GO nanosheets. For example, the presence of the solder peak in the Ni 2p spectra (supplementary Figure 4c) at ~862 eV clearly indicates the presence of the Ni(OH)₂ (ref 26).

4. Multivalent cations are known to form cross-linking with the functional groups of GO. The selectivity of the Ni²⁺ towards the GO and pPD should be studied, which is missing this work. They should compare the XPS and the IR spectra of the pure GO and GO treated with Ni(NO₃)₂ and NH₄OH, pure Ni-pPD metal complex and the Ni-pPD@rGO nanosheets.

5. How did the authors separate the Ni-pPD@rGO nanosheets from the bi-products formed during the reaction? The XPS analysis indicates the obtained Ni-pPD@rGO nanosheets are not in the pure form. Rather it is a mixture with Ni(OH)₂.

The reaction of Ni²⁺ ions with base forms Ni(OH)₂ nanosheets, which are insoluble in water and have positive zeta potential, which will lead to a decrease in the zeta potential.

6. As claimed in Figure 1h, the d-spacing of the Ni-pPD@rGO is 9.6 Å, which gives a free space of ~ 6.2 Å (subtracting the basal plan thickness 3.4 Å from d-spacing). Now, it is not clear how two Ni-pPD sheets each of 2.6 Å will fit between two GO flakes when the space between a GO and the Ni-pPD is 3.63 Å (Figure 1d). Even neglecting the space between two Ni-pPD monolayers, the total free space between two GO layers should be 12.46 Å.

7. "GOMs normally present only one X-ray diffraction (XRD) peak at around 11° (~0.8 nm) as shown in Fig. 1i." On careful observation, the peak for GOM is centered at around 10°-10.4° (0.88 to 0.85 nm).

Similarly for MOGM, the d-spacing is mentioned as 0.96nm, but the peak 002 is centered at around 9.5°-9.8° resulting in a d-spacing of around 0.93 to 0.9 nm.

Overall, the manuscript lacks proper characterization of the prepared samples, mismatching the results discussed with the corresponding Figure.

Reviewer #2 (Remarks to the Author):

This manuscript titled " Anomalous Liquid Gating from Atomic-scale Graphene Capillaries for Precise and Ultrafast Molecular Sieving" presents anomalous liquid gating from nanofiltration

membranes based on a graphene nanoconfinement effect, fabricated from spontaneous restacking of island-on-nanosheet microstructure. The graphene nanoconfinement facilitates high-speed, diffusion-free water transport in a way analogous to Newton's cradle-like Grotthus conduction. These experiments are rationalized, and the results are sufficient to support the conclusion. Before considering this manuscript for publication, the authors should consider the following points in any revision as follows:

1. The authors state nanoscale confined water structures have solid-like properties that hinder diffusion under osmosis. It is not the liquid we usually know. Basically, liquid gating technology is to use the liquid as a mechanical gate to control the transport of fluid. Therefore, the liquid gating phenomenon described in this manuscript is not appropriate. The title needs to be reconsidered. For example, using "water molecular gating".
2. The introduction is not well arranged, and the logic is not clear. And the use of many conjunctions is confusing. For example, there is no causal relationship between "As such, ineffective ion sieving properties have been found on reduced graphene oxide membranes" and the previous sentence.
3. AFM images of the nanosheet thickness before restacking are encouraged to be provided for better comparison.
4. Although the optimized structure is shown in the paper (Fig. 1d), it is necessary to give the initial structure before optimization in SI.
5. For Fig. 1h, please explain further why does the restacked structure provide distinct free space is provided? And mark the area corresponding to free space in the figure.
6. In the sentence "We examined an unexpected permeation behavior on MOGMs (Fig. 2a)", the Fig. 2a did not show the unexpected behavior. And abbreviations should be defined when they first appear in the text. Write the full name and give the abbreviation in parentheses.
7. The caption of Fig. 2c was unclear, does the upper subgraph in Fig. 2c represent microscopic states at different times? And the explanation for Figure 2c is presented in an untimely manner, appearing only in the concluding discussion section. This layout lacks coherence and may not be considered optimal.
8. In the sentence "The membranes made of the restacked Ni-pPD@rGO nanosheets have then been fabricated by vacuum filtration, followed by the controlled evaporation under external pressures.", What does it mean to control evaporation under external pressure? Does it have any special effect to the fabricating process?
9. How durable and resistant to staining is the membrane?
10. The contact angle of the MOGMs is 84°. How does its hydrophilic or hydrophobic nature affect the anomalous gating behavior?
11. The water flux keeps increasing up to 2.5 L m⁻² h⁻¹ at 9.8×10⁻³ bar ("ON" state in Fig. 2b). If the hydrostatic pressure is continuously increased, will the water flux rate continue to increase? Is there a turning point hydrostatic pressure?
12. This work is interesting, but it lacks an appropriate schematic diagram to clearly explain its mechanism, making it difficult for readers to understand easily. It is better to modify Fig. 2 to help reader better understand the mechanism of the anomalous gating behavior.
13. For the DFT simulation, what are the criteria for hydrogen bonding between water molecules? In addition, the role of Ni-pPD is not reflected in the model. Please explain in detail the special role of Ni-pPD in this system and the significant difference in properties between rGO nanosheets without Ni-pPD.
14. In the sentence "the energy barrier was obtained by subtracting the energy before the move from that after the move" of supplementary Note3, does the energy mean all energy in the system?
15. For Fig. S20, is there any quantitative correlation between the On/Off Switch Height and the hydrating ion radius of different ions?
16. Please change the term "Van der Waals" to "van der Waals".
17. According to the theoretical computations, the "ON" state shows the bulk movement of water molecules, which seems not the same as the movement of Newton's cradle-like Grotthus conduction. Authors should consider whether this description is appropriate or not.
18. The shaded and solid sections of the bar chart in Fig 4c should indicate the meaning clearly.
19. How does the graphene capillaries size influence ion diffusion and water transport.
20. The authors are suggested to check through the manuscript to improve the English. It is difficult to read.
21. Many recent publications related to ion transport behavior, such as Nat Commun 2022, 13,

6709, Nano Letters 2020, 20: 6937–6946, Joule 2023, 7: 251–253. In order to help readers better understand the importance of this work, these references should be cited.

I will be happy to recommend for publication a revised version of the manuscript in Nature communications.

Reviewer #3 (Remarks to the Author):

This manuscript describes a clever catalytic synthesis method to fabricate a sub-nano-confined, graphene-based microstructure. Ions can be selectively sieved by gating water molecules on/off. The mechanism is presented that water forms crystal-like water nanosheets inside the graphene capillaries and the application of a small hydrostatic pressure causes the water nanosheet to slide as a whole, enabling fast transport. This is very interesting. In addition, combining the experimental data with MD simulation has greatly improved the quality of this work. The manuscript can be considered for publications after minor revisions, as the comments listed below.

1. How to make sure there is only one layer of Ni-pPD on top of rGO in assisting the formation of sub-nano-confined graphene capillaries?
2. The process of ON/OFF is unclear. The authors should clearly define how the pressure is applied.
3. The membrane is not a single capillary, how to rationalize the simulation with only one graphene capillary without considering the tortuous pathways along ~2.3-micron thick graphene membranes?
4. The resolution in several figures is not high enough. For example, Fig. 1b
5. How is the comparison between the DFT simulation results and experimental results?
6. The flux data on Figure S13 extrapolate to zero LMH at zero applied hydrostatic pressure, while for the data on the Figure S12, the data extrapolate to about 4LMH. What is the reason for such a difference?
7. The authors refer to the nanosheets as graphene at times which is not the correct use of terminology. The authors should choose and use the appropriate term.
8. There are several errors in references formats.

RESPONSE TO REVIEWERS' COMMENTS

Reviewer #1:

The manuscript entitled “Anomalous Liquid Gating from Atomic-scale Graphene Capillaries for Precise and Ultrafast Molecular Sieving” has many issues which need to be rectified before submitting it to any scientific journal. At the current stage, it is not suitable for publication in Nature Communication. Some (not all) of the major issues are listed below.

Reply and revision: We greatly appreciate the valuable comments and all the very helpful suggestions from the reviewer.

1. The claim “for the first time, allows us to take full advantage of atomic-scale, graphene capillaries under osmosis at room temperature” is misleading. Firstly, the manuscript is on reduced graphene oxide capillaries. Secondly, one of the authors Prof Lei Jiang group are already pioneer in the power generation from GO membranes, including osmosis. Hence the claim is inappropriate.

Reply and revision: Thank you for the referee’s valuable comments. Both oxidized graphene capillaries and pristine graphene capillaries are present in GO membranes (GOMs) and rGO-based membranes, as shown in **Figure R1a**. It is noted that **there are no capillaries specifically designated as reduced graphene oxide (rGO) capillaries; instead, rGO-based membranes contain a higher number of pristine graphene capillaries than those in GOMs.** ^[R1]

Figure R1. (a) Illustration of oxidized graphene capillaries and pristine graphene capillaries in GOMs and rGO-based membranes. (b) Reduced graphene region tends to restack, sealing the atomic-scale, graphene capillaries.

Moreover, although pristine graphene capillaries are present in both types of membranes, **the advantages of atomic-scale, graphene capillaries have not been fully explored under osmosis at room temperature.** As mentioned by the referee, GOMs have been tested for water-ion separation under osmosis. Nevertheless, GOMs exhibit a high hydrophilicity due to a significant presence of oxygenated functional groups (40-50% of the oxidized region).^[R2] This abundance of functional groups can hinder the rate of water transport, thereby obscuring the

beneficial effects of atomic-scale graphene capillaries under osmosis at room temperature. On the other hand, rGO-based membranes contain significantly fewer oxygenated functional groups as compared to GOMs, resulting in a higher proportion of pristine graphene capillaries. However, the challenge for rGO-based membranes is the tendency of the reduced region to restack, effectively sealing the atomic-scale, graphene capillaries (**Figure R1b**), which restricts permeation only through large graphitic pores and deteriorates the ion sieving properties. Despite demonstrating fast frictionless flow of water, rGO-based membranes do not exhibit as efficient ion sieving as GOMs. **Therefore, the main challenge lies in the control of 2D graphene slit pore size with angstrom precision in rGO-based membranes.** In this work, we have successfully achieved this goal, resulting in the simultaneous achievement of ultrafast bulk water flow of 2.5-45.4 L m⁻² h⁻¹ and negligible ion permeation rates of ~10⁻⁴ mol m⁻² h⁻¹ for various alkali-metal ions. These remarkable properties are governed by a newly-discovered, anomalous water molecular gating mechanism.

Therefore, as suggested by the referee and to provide greater precision, we have revised the relevant description in our revised manuscript: “**For the first time, it enables us to fully exploit the potential of atomic-scale graphene capillaries in rGO-based membranes under osmosis at room temperature.**” In addition, we would like to clarify that our membranes are rGO-based membranes rather than GO-based membranes. The most direct evidence supporting this is the contact angle results shown in Fig. 2e. The contact angle of our membranes is 84°, which is very close to that of pure graphite (85°).^[R3] In contrast, GO-based membranes are hydrophilic with a contact angle of approximately 44.5°. Further details regarding this distinction have been included in our responses to the referee’s next two questions.

Accordingly, we revised the term “graphene-based nanofiltration (NF) membranes” to “rGO-based NF membranes” in the revised manuscript. In addition, we revised the relevant description on Page 4 of the Revised Manuscript as follows:

“For the first time, it enables us to fully exploit the potential of atomic-scale graphene capillaries in rGO-based membranes under osmosis at room temperature.”

2. It is quite surprising to know that the oxygenated functional groups disappear in the resulting nanosheets. GO nanosheets are known to have different oxygen-containing functional groups like –O–, –OH, –COOH, C=O, etc. It is very hard to reduce all these functional groups especially the –O– and –COOH even in harsh conditions. In the FTIR compared in the supplementary Figure 3, the authors have indicated the presence of the OH and the C-OH groups in the Ni-pPD@rGO nanosheets, which directly contradicts their claim of “the oxygenated functional groups disappear in the resulted nanosheets, implying the reduction of GO to rGO after the reaction.” Careful observation in the spectra also indicate the presence of other oxygen functional groups like C-O-C in the Ni-pPD@rGO nanosheets. In addition, the XPS (supplementary Figure 4a) spectra of the Ni-pPD@rGO nanosheets have a very strong

peak for O 1s at ~530 eV. This clearly indicates the presence of different oxygen functional groups.

Reply and revision: Thanks for the referee’s critical comments. We acknowledge the referee's interpretation and agree that Supplementary Figs 3 and 4a provide clear evidence of the presence of distinct oxygen functional groups on our Ni-pPD@rGO nanosheets. This is understandable because even after the reduction of GO to rGO, a small amount of oxygenated functional groups may still remain, as observed in other GO reduction methods.^[R4] Thus, the term “disappear” could potentially confuse the referee and potential readers. Therefore, we have revised the manuscript to clarify that “the oxygenated functional groups **have been substantially removed in the resulted nanosheets**, implying the reduction of GO to rGO after the reaction.”

Although different oxygen functional groups are present, the quantity of oxygenated functional groups in our Ni-pPD@rGO nanosheets has substantially decreased, compared to the original GO nanosheets. This reduction has also been confirmed through both FTIR and XPS analyses, as shown in Supplementary Figs 3, 4a, and 4b in the revised manuscript. Notably, in Supplementary Fig. 4b (also depicted in **Figure R2a**), the C 1s XPS spectrum of GO is compared with that of Ni-pPD@rGO nanosheets, revealing a substantial suppression of the oxygenated carbon peak at approximately 286.8 eV after the proposed reaction. This clear reduction in intensity indicates the removal of oxygenated functional groups from carbon to a significant degree. Moreover, the contact angle results depicted in Fig. 2e of the revised manuscript provide direct evidence of GO reduction. The contact angle of Ni-pPD@rGO membranes is 84°, which is very close to that of pure graphite (85°).^[R3] In contrast, GO-based membranes are hydrophilic with a contact angle of about 44.5° (**Figure R2b**).

Figure R2. (a) XPS analyses of Ni-pPD@rGO nanosheets in comparison with GO nanosheets. (b) Contact angle and XRD analyses. (c) Pore size distribution analysis.

More importantly, the presence of a small amount of oxygenated functional groups on rGO does not significantly impact the discussion and conclusions presented in our manuscript. **The reduction of GO to rGO does not imply that all capillaries become reduced graphene oxide capillaries. Rather, it signifies a substantial decrease in the percentage of oxidized graphene**

capillaries, resulting in a larger proportion of exposed pristine graphene regions. However, at the atomic scale, these pristine graphene capillaries tend to become sealed because of restacking phenomenon (**Figure R1b**), as reported in previous studies.^[R4] To address this challenge, we developed a novel method to form a Ni-pPD island-on-rGO nanosheet microstructure. The spontaneous restacking of Ni-pPD@rGO nanosheets leads to the creation of stable atomic-scale pristine graphene capillaries, predominantly with a size of 0.6 nm. This result has also been experimentally confirmed through pore size distribution analyses, as shown in Fig. 2f (**Figure R2c**). Therefore, the sub-nano graphene pore structure in our membranes exhibits similar characteristics to those of 2D molecular sieves, demonstrating the delicacy of this synthesis strategy and supporting our discussion regarding the anomalous water molecular gating mechanism.

Accordingly, we changed our claim to be “the oxygenated functional groups have been substantially removed in the resulted nanosheets, implying the reduction of GO to rGO after the reaction.” on **Page 6 of the revised manuscript.**

3. How did the authors confirm the formation of the Ni-pPD complex? They did not discuss it properly in any of the characterization techniques. For example, in the FTIR, which peak confirms the Ni-N bond (N=Q=N peak?). In XPS, the peak under O 1s at ~530 eV is not well described.

In all the characterization techniques, the authors compared the spectra of the GO and Ni-pPD@rGO nanosheets. Although the formation of the Ni-pPD complex is a known reaction (as mentioned in the ref 26), but this formation might be different in the presence of the GO nanosheets. For example, the presence of the solder peak in the Ni 2p spectra (supplementary Figure 4c) at ~862 eV clearly indicates the presence of the Ni(OH)₂ (ref 26).

Reply and revision: Thank you for the valuable comments. The Ni-pPD complex possesses distinct functional groups, including -C-N groups, -C=N groups, N=Q=N groups (Q representing the quinonoid ring), -NH-, -N=, and Ni-N. As also raised by the referee, N=Q=N groups are a particular substructure in Ni-pPD complex formed through the designated catalytic reaction, according to ref. 29 (previous ref. 26).

In order to confirm the formation of the Ni-pPD complex, we have conducted EDX mapping, FTIR and XPS analyses in this work. Firstly, STEM-EDX mapping images in Fig. 1b and Supplementary Fig. 2 show homogeneous distribution of N on the resulted nanosheets. Secondly, the characteristic peaks in FTIR Spectra (Supplementary Fig. 3) at 1274 cm⁻¹ and 1168 cm⁻¹ correspond to -C-N and N=Q=N groups, respectively. Furthermore, XPS analyses (Supplementary Fig. 4) also show the characteristic peaks of Ni-pPD (-C-N and -C=N groups in C 1s, -NH-, -N=, and Ni-N bonds in N 1s), consistent with the observations found in the FTIR results. **In particular, the presence of N=Q=N groups in the FTIR spectra and Ni-N bonds in the XPS spectra provides strong evidence for the formation of the Ni-pPD complex.**

The comparison of XPS and FTIR spectra of pure GO, pure Ni-pPD metal complex, GO treated with Ni(NO₃)₂ and NH₄OH (without adding pPD), and the Ni-pPD@rGO nanosheets (with the addition of pPD) has been added to **Supplementary Figs 3-5 in the revised manuscript**. These additional results serve to further validate the formation of the Ni-pPD complex. Please refer to our responses to Question 4 raised by the referee for more details.

In XPS analysis, the comparison of O 1s spectra at ~530 eV for GO and Ni-pPD@rGO has been added, as shown in the following figure (**Figure R3**), which also clearly demonstrates the reduction of GO to rGO.

Figure R3. O1s XPS spectrum of Ni-pPD@rGO nanosheets in comparison with that of GO nanosheets.

We concur with the referee's observation that there may be traces of Ni(OH)₂ in our samples. However, the predominant component is the 2D Ni-pPD complex. If Ni(OH)₂ was the primary component, the entire sample would still have hydrophilic characteristics similar to GO nanosheets.^[R5] However, Fig. 2e shows that the contact angle of our samples is 84°, indicating that the main component is the 2D Ni-pPD complex on rGO nanosheets. More discussion about Ni(OH)₂ has also been provided in our responses to Question 5 raised by the referee.

Accordingly, the comparison of O 1s spectra at ~530 eV for GO and Ni-pPD@rGO have been added in Supplementary Fig. 4c in the revised manuscript.

4. Multivalent cations are known to form cross-linking with the functional groups of GO. The selectivity of the Ni²⁺ towards the GO and pPD should be studied, which is missing in this work. They should compare the XPS and the IR spectra of the pure GO and GO treated with Ni(NO₃)₂ and NH₄OH, pure Ni-pPD metal complex and the Ni-pPD@rGO nanosheets.

Reply and revision: We agree with the referee's valuable comments. The first synthesis step was to disperse GO nanosheets within Ni^{2+} solutions while expelling dissolved oxygen in water. Thus, Ni^{2+} ions would be adsorbed onto the oxygenated functional groups on GO nanosheets in this step. Then, de-oxygenated pPD solutions were added together with NH_4OH to initiate the catalytic reaction between the adsorbed Ni^{2+} and pPD on top of the oxygenated functional groups. It is noted that the reaction of forming Ni-pPD can only occur at the presence of O, according to ref. 29 (previous ref. 26). **Therefore, the cross-linking effect mentioned by the referee actually played a significant role in the synthesis of Ni-pPD@rGO nanosheets. In other words, the complex of the Ni^{2+} towards GO and pPD are not two competing processes, but interactive and sequential processes.**

As suggested by the referee, we added more characterization results to compare four different types of samples, (1) pure GO, (2) GO treated with $\text{Ni}(\text{NO}_3)_2$ and NH_4OH (without adding pPD), (3) Ni-pPD@rGO nanosheets (with the addition of pPD), and (4) pure Ni-pPD metal complex. **Figure R4** of FTIR analyses show that Sample (2) of GO treated with $\text{Ni}(\text{NO}_3)_2$ and NH_4OH (without adding pPD) almost maintain the functionality of GO. No reduction of GO occurred. More importantly, N=Q=N groups, the most characteristic feature of Ni-pPD complex, are present only for Sample (3) of Ni-pPD@rGO and Sample (4) of pure Ni-pPD in FTIR spectra. **All these results suggest the participation of pPD in the reaction for the formation of Ni-pPD complex, and at the same time, for the reduction of GO to rGO.**

Figure R4. FTIR spectra of GO, $\text{Ni}(\text{OH})_2$ @GO without adding pPD, Ni-pPD@rGO nanosheets with adding pPD, pure Ni-pPD.

In **Figure R5** of XPS analysis, it is also demonstrated that, without the addition of pPD, no GO reduction occurred for Sample (2).

Figure R5. (a) C1s and (b) O1s XPS spectra of Ni-pPD@rGO nanosheets (with adding pPD) and Ni(OH)₂@GO (without adding pPD). (c) Comparison of Ni 1s for pure Ni-pPD, Ni-pPD@rGO nanosheets (with adding pPD) and Ni(OH)₂@GO (without adding pPD).

All in all, the conclusion is much clearer in the revised manuscript, thanks to the referee's valuable comment.

Accordingly, the following description was added in Method section on Page 21 of the revised manuscript,

“For comparison, Ni(OH)₂@GO nanosheets were also synthesized by treating GO only with Ni(NO₃)₂ and NH₄OH (without adding the pPD solution).”

In addition, the XPS and FTIR analyses of GO treated only with Ni(NO₃)₂ and NH₄OH, and pure Ni-pPD metal complex have been added in Supplementary Figs 3-5 in the revised manuscript.

5. How did the authors separate the Ni-pPD@rGO nanosheets from the bi-products formed during the reaction? The XPS analysis indicates the obtained Ni-pPD@rGO nanosheets are not in the pure form. Rather it is a mixture with Ni(OH)₂.

The reaction of Ni²⁺ ions with base forms Ni(OH)₂ nanosheets, which are insoluble in water and have positive zeta potential, which will lead to a decrease in the zeta potential.

Reply and revision: Thank you for the referee's critical comments. After the synthesis, the resulted black powders were filtered and then washed with Milli-Q water and ethanol thoroughly, in order to separate the Ni-pPD@rGO nanosheets from the possible bi-products formed during the reaction.

We concur with the referee's observation that there may be traces of Ni(OH)₂ in our samples. However, the predominant component is the 2D Ni-pPD complex. If Ni(OH)₂ was the primary component, the entire sample would still hydrophilic characteristics similar to GO

nanosheets.^[R5] However, Fig. 2e shows that the contact angle of our samples is 84°, indicating that the main component is the 2D Ni-pPD complex on rGO nanosheets. Figures R2-R7 also demonstrate that the main component is 2D Ni-pPD complex on top of rGO nanosheets.

Figure R8. Illustration of the possible defects as few oxygenated functional groups and Ni(OH)₂ in MOGMs, which would not affect water/ion permeation behavior as long as the membrane is thick enough.

Despite the presence of remaining oxygenated functional groups and Ni(OH)₂, they can be regarded as defective and ancillary structures within our membrane samples. The primary structures in the resulting Ni-pPD@rGO membranes remain as atomic-scale graphene capillaries. As long as the membrane is adequately thick, these defective structures cannot connect to form a continuous pathway. **Consequently, the atomic-scale graphene capillaries serve as the rate-limiting factor for water and ion transport across the membrane (Figure R8).** Hence, the existence of these defective structures within the membrane does not impact the discussion and conclusions presented in this study.

Accordingly, the following description was added as Supplementary Note 7 in Supplementary Information on Page 26 of the revised manuscript,

“Although there may be remaining oxygenated functional groups on rGO and traces of Ni(OH)₂ present, they can be considered as defective and secondary structures within MOGMs. The primary structures in MOGMs remain as atomic-scale graphene capillaries. As long as the membrane is sufficiently thick, these defective structures cannot form a continuous pathway. Therefore, the atomic-scale graphene capillaries continue to serve as the rate-limiting factor for water and ion transport across the membrane.”

6. As claimed in Figure 1h, the d -spacing of the Ni-pPD@rGO is 9.6 Å, which gives a free space of ~ 6.2 Å (subtracting the basal plan thickness 3.4 Å from d -spacing). Now, it is not clear how two Ni-pPD sheets each of 2.6 Å will fit between two GO flakes when the space between a GO and the Ni-pPD is 3.63 Å (Figure 1d). Even neglecting the space between two Ni-pPD monolayers, the total free space between two GO layers should be 12.46 Å.

Reply and revision: Thank you for the referee’s critical comments. **Fig. 1e (previous Fig. 1d) may be a little misleading. 3.63 Å in Fig. 1e is actually the d -spacing between the rGO and the restacked Ni-pPD complex, not the free space.** This is understandable, because it is the normal π - π stacking. Therefore, three layers’ π - π stacking, including rGO-pPD, pPD-pPD, and pPD-rGO, should result in the d -spacing of the restacked Ni-pPD@rGO being around $3 \times 3.63 = 10.9$ Å (**Figure R9a**). Furthermore, as also shown in Figure R9a, because of small interaction between the 1st-layer rGO and 3rd-layer pPD, rGO and pPD nanosheets will be deformed a little bit, contracting the whole d -spacing to be 9.6 Å. In this way, the free space between the 1st-layer rGO and 3rd-layer pPD is 6.01 Å $- 3.4$ Å = 2.61 Å, and that between the 1st-layer rGO and 4th-layer rGO is 9.58 Å $- 3.4$ Å = 6.18 Å (**Figure R9b**).

Figure R9. Optimized structure of restacked Ni-pPD@rGO nanosheets computed by DFT. **a**, Ni-pPD@rGO nanosheets before and after optimization. All values are d -spacings, not free space. **b**, The restacking results in the free space of 2.6 Å and 6.2 Å for molecules to transport.

Accordingly, we have clarified that 3.63 Å is the d -spacing between the rGO and the Ni-pPD monolayer island in Fig. 1e (previous Fig. 1d) in the revised manuscript.

7. “GOMs normally present only one X-ray diffraction (XRD) peak at around 11° (~0.8 nm) as shown in Fig. 1i.” On careful observation, the peak for GOM is centered at around 10° - 10.4° (0.88 to 0.85 nm).

Similarly for MOGM, the *d*-spacing is mentioned as 0.96nm, but the peak 002 is centered at around 9.5°-9.8° resulting in a *d*-spacing of around 0.93 to 0.9 nm.

Reply and revision: Thanks for the referee's careful observations. We corrected our claims to be "GOMs normally present only one X-ray diffraction (XRD) peak at around 10.4° (~0.85 nm) as shown in Fig. 2e." and "The XRD result obtained for MOGMs, however, demonstrates four distinct peaks of (002), (004), (006), and (008), with the interspacing of (002) being 0.93 nm and that of (006) being 0.32 nm, strongly supporting our conclusions about the restacked repeating unit structure of Ni-pPD@rGO nanosheets" on Page 10 in the revised manuscript.

Please note that the XRD analysis of GOMs is also dependent on the relative humidity of the environment, so that it may cause some variations. Moreover, there are four distinct peaks of (002), (004), (006), and (008) for MOGMs. We can also calculate *d*-spacings of (002) from *d*-spacings of (004), (006), and (008), which are 0.93 nm, 0.97 nm, and 0.94 nm. This XRD result clearly suggests the superlattice structure for MOGMs.

Reviewer #2:

This manuscript titled " Anomalous Liquid Gating from Atomic-scale Graphene Capillaries for Precise and Ultrafast Molecular Sieving" presents anomalous liquid gating from nanofiltration membranes based on a graphene nanoconfinement effect, fabricated from spontaneous restacking of island-on-nanosheet microstructure. The graphene nanoconfinement facilitates high-speed, diffusion-free water transport in a way analogous to Newton's cradle-like Grotthus conduction. These experiments are rationalized, and the results are sufficient to support the conclusion. Before considering this manuscript for publication, the authors should consider the following points in any revision as follows:

I will be happy to recommend for publication a revised version of the manuscript in Nature communications.

Reply and revision: We greatly appreciate the positive comments and all the very helpful suggestions from the reviewer.

1. The authors state nanoscale confined water structures have solid-like properties that hinder diffusion under osmosis. It is not the liquid we usually know. Basically, liquid gating technology is to use the liquid as a mechanical gate to control the transport of fluid. Therefore, the liquid gating phenomenon described in this manuscript is not appropriate. The title needs to be reconsidered. For example, using "water molecular gating".

Reply and revision: We greatly appreciate the referee's important comment and helpful suggestion. The phrase "water molecular gating" indeed grasps the essence of this work. Thus, following the suggestion, we have changed the phrase of "liquid gating" to "water molecular gating" in the revised manuscript.

2. The introduction is not well arranged, and the logic is not clear. And the use of many conjunctions is confusing. For example, there is no causal relationship between " As such, ineffective ion sieving properties have been found on reduced graphene oxide membranes" and the previous sentence.

Reply and revision: We really appreciate the referee's valuable comments. Following the helpful suggestion, we have reorganized the introduction in the revised manuscript. We hope this version would be much easier for readers to understand.

3. AFM images of the nanosheet thickness before restacking are encouraged to be provided for better comparison.

Reply and revision: We really appreciate the referee for raising this critical and valuable comment. Unfortunately, we cannot obtain the nanosheet before restacking. It is the spontaneous restacking after the catalytic reaction, similar to the spontaneous restacking of rGO nanosheets after GO reduction. However, the referee indeed reminds us to check the fine structure of Ni-pPD monolayers on top of the restacked Ni-pPD@rGO nanosheets. Please see the following

figure (Figure R10), which is consistent with HAADF image in Fig. 1c in the revised manuscript.

Figure R10. AFM analysis showing monolayer island structure on top of the spontaneously restacked Ni-pPD@rGO nanosheets. **a**, AFM image. **b**, Height profile. The fine monolayer island structure has been highlighted by pink circles.

Accordingly, the above figure has been added as Supplementary Figure 8 in the revised manuscript.

4. Although the optimized structure is shown in the paper (Fig. 1d), it is necessary to give the initial structure before optimization in SI.

Reply and revision: We really appreciate the referee's comments. The initial structure has been provided in the revised manuscript, as also shown below.

Figure R11. The initial structure of Ni-pPD@rGO nanosheets before relaxation. **a,b**, view from two different angles.

Accordingly, the above figure has been added as Supplementary Figure 7 in the revised manuscript. In addition, the following description has been added on Page 6 of the revised manuscript,

“As compared with the heterostructure before relaxation (Supplementary Fig. 7), Fig. 1e demonstrates that the four benzene rings around Ni ions in Ni-pPD tend to be planarized by rGO through π - π stacking.”

5. For Fig. 1h, please explain further why does the restacked structure provide distinct free space is provided? And mark the area corresponding to free space in the figure.

Reply and revision: We really appreciate the referee to point this out. As shown in **Figure R12a,b** (Supplementary Fig. 9 in the revised manuscript), the d -spacing between the 1st-layer rGO and 3rd-layer pPD is 6.0 Å, giving the free space of 2.6 Å; and also, the d -spacing between the 1st-layer rGO and 4th-layer rGO is 9.6 Å, giving the free space of 6.2 Å. The area corresponding to free space has been marked in Fig. 2d in the revised manuscript, which is also shown in **Figure R12c**.

Optimized Restacked Structure

Figure R12. Optimized structure of restacked Ni-pPD@rGO nanosheets computed by DFT. a, All labels are interlayer spacings. **b,c** The restacking results in the free space of 2.6 Å and 6.2 Å for molecules to transport.

Accordingly, Fig. 2d (previous Fig. 1h) has been redrawn in the revised manuscript. In addition, the following descriptions have been added on Page 8 in the revised manuscript,

“As such, the d -spacing between the 1st-layer rGO and 3rd-layer pPD was calculated to be 6.0 Å, giving the free space of 2.6 Å (marked as pink slash lines in Fig. 2d); and also, the d -spacing between the 1st-layer rGO and 4th-layer rGO was determined to be 9.6 Å, giving the free space of 6.2 Å (marked as blue slash lines in Fig. 2d).”

6. In the sentence “We examined an unexpected permeation behavior on MOGMs (Fig. 2a)”, the Fig. 2a did not show the unexpected behavior. And abbreviations should be defined when they first appear in the text. Write the full name and give the abbreviation in parentheses.

Reply and revision: We really appreciate the referee to point this out. The sentence has been corrected on Page 11 of the revised manuscript to be,

“We examined an unexpected permeation behavior on MOGMs (Fig. 2b).”

In addition, we corrected the definition of the abbreviation on Page 9 of the revised manuscript to be

“These new membranes are, therefore, denoted as metal-organic (Ni-pPD) pillared, graphene-based membranes (MOGMs).”

7. The caption of Fig. 2c was unclear, does the upper subgraph in Fig. 2c represent microscopic states at different times? And the explanation for Figure 2c is presented in an untimely manner, appearing only in the concluding discussion section. This layout lacks coherence and may not be considered optimal.

Reply and revision: We agree with the referee’s valuable comments. According to the referee’s suggestion, we have moved Fig. 2c and the corresponding discussion to the end of the revised manuscript. In addition, Fig. 2c has been modified to be a new figure of Fig. 6 in the revised manuscript, which was shown as **Figure R13** in our responses to Question 12.

8. In the sentence “The membranes made of the restacked Ni-pPD@rGO nanosheets have then been fabricated by vacuum filtration, followed by the controlled evaporation under external pressures.”, What does it mean to control evaporation under external pressure? Does it have any special effect to the fabricating process?

Reply and revision: We really appreciate the referee to point this out. As described in Methods and also illustrated in Supplementary Figure 10, after vacuum filtration, the membranes were immediately clamped between two glass plates and dried under vacuum at 70°C. The control of evaporation is an important step for the synthesis of MOGMs with well-defined, atomic-scale graphene capillaries at 6 Å. Too fast or uneven evaporation will cause the formation of large pores, in which case the anomalous water molecular gating mechanism would not be observed.

To better clarify this point, the following description has been added to the Method section on **Page 22 of the revised manuscript**,

“The controlled evaporation is an essential step to make sure MOGMs with well-defined, atomic-scale graphene capillaries at 6 Å.”

9. How durable and resistant to staining is the membrane?

Reply and revision: We really appreciate the referee's valuable comments. The referee raised a very critical comment. This study is mainly to discover the anomalous water molecular gating behavior and its underlying mechanism. In addition, the durability and resistance to staining can also be improved by the design of a composite membrane based on the current MOGMs. Thus, we intend to explore this avenue in our future research and hope to report on it in the near future.

10. The contact angle of the MOGMs is 84°. How does its hydrophilic or hydrophobic nature affect the anomalous gating behavior?

Reply and revision: We really appreciate the referee's valuable comments. The hydrophilic or hydrophobic nature of the atomic-scale capillaries is essential in activating the anomalous water molecular gating behavior. The presence of oxygenated functional groups would disturb the formation of 2D crystal-like, hydrogen-bonded water network within the atomic-scale capillaries. Thus, this anomalous permeation phenomenon has not been observed on hydrophilic GOMs in previous reports. On the other hand, we believe that the atomic-scale hydrophobic capillaries (contact angle $> 90^\circ$) would not generate this anomalous permeation phenomenon either, since it is difficult for water molecules to enter the hydrophobic capillaries. Therefore, atomic-scale, graphene capillaries is the key for the anomalous gating behavior.

According to our calculations, the fundamental reason is the attractive van der Waals interactions between the graphene walls and water molecules. Although it is relatively weak in large capillaries, this short-range attraction plays a key role within the atomic-scale graphene capillaries and a large capillary pressure can be generated (in the order of 1,000 bar). Therefore, the contact angle of 84° for MOGMs, which is close to the contact angle of water on graphite (85°),^[R3] is one of the most important characteristics in activating the anomalous water molecular gating behavior reported in this work.

11. The water flux keeps increasing up to $2.5 \text{ L m}^{-2} \text{ h}^{-1}$ at $9.8 \times 10^{-3} \text{ bar}$ ("ON" state in Fig. 2b). If the hydrostatic pressure is continuously increased, will the water flux rate continue to increase? Is there a turning point hydrostatic pressure?

Reply and revision: We really appreciate the referee's valuable comments. As displayed in Fig. 5d in the revised manuscript, the water flux rate will continue to increase if the hydrostatic pressure increases further. In addition, the referee raised a very interesting topic about the turning point for the continuous increase of the hydrostatic pressure. It requires the systematic investigation about the extremes of the newly-found anomalous permeation phenomenon. This work is undertaken and we hope the related work will be reported in the near future.

12. This work is interesting, but it lacks an appropriate schematic diagram to clearly explain its mechanism, making it difficult for readers to understand easily. It is better to modify Fig. 2 to help reader better understand the mechanism of the anomalous gating behavior.

Reply and revision: We really appreciate the referee's valuable comments. We have modified Fig. 2c and moved it to a new figure of Fig. 6 at the end of the revised manuscript. In Fig. 6 (also

shown in **Figure R13**), schematic diagrams have been demonstrated to clearly explain its mechanism; especially, Newton’s cradle-like Grotthus Conduction for fast water flux.

Figure R13. Microscopic illustrations of water molecular gating mechanism. a, “OFF” state. b, “ON” state with Newton’s cradle-like Grotthus Conduction originated from highly-connected hydrogen-bonded network.

The new Fig. 6b (previous Fig. 2c) represent two continuous states for the collision of one extra molecule onto the 2D ordered water nanosheets, leading to almost instant transport of the water molecule to the other side of the capillary. This is analogous to Newton’s cradle-like Grotthus Conduction. **More importantly, the two continuous states in Fig. 6b together represent only one snapshot for each collision.** In real operations, it should be continuous collision of water molecules from the left, resulting in instant but continuous water transport to the other side of the capillary. **Thus, the continuous mode of this analogy forms bulk flow of water.**

13. For the DFT simulation, what are the criteria for hydrogen bonding between water molecules? In addition, the role of Ni-pPD is not reflected in the model. Please explain in detail the special role of Ni-pPD in this system and the significant difference in properties between rGO nanosheets without Ni-pPD.

Reply and revision: We really appreciate the referee’s valuable comments. As inspired by the referee, we put the quantitative analysis of the average number of hydrogen bonds in the revised manuscript in Supplementary Fig. 20.

As well accepted in the literature, the definition for a hydrogen-bond is that the distance between two oxygen atoms $R_{o-o} < 3.5 \text{ \AA}$ and $\alpha < 30^\circ$.^[R6] As shown in Figure R14, putting these criteria into VMD software, the average number of hydrogen bonds for the structure in **a** can be

calculated to be 44 in **b**. Considering the hexagonal type of unit cell, there are around 18.2 water molecules in **a**. In addition, each hydrogen bond is shared by two water molecules, and thus, there are averagely 4.83 hydrogen bonds around each water molecule ($\text{HB} \cdot \text{H}_2\text{O} = 4.83$).

Figure R14. Quantitative determination of the average number of hydrogen bonds around each water molecule. **a**, The water structure within the atomic-scale graphene capillaries. **b**, The calculation of the average number of hydrogen bonds in **a** using VMD software.

Ni-pPD monolayer islands were used as pillars between two rGO nanosheets to create atomic-scale graphene capillaries. If there is no Ni-pPD, two rGO nanosheets would restack and seal the atomic-scale graphene capillaries. In addition, the Ni-pPD monolayer islands is mainly composed of the graphitic structure. Thus, the role of Ni-pPD is not reflected in the model.

Accordingly, the above figure has been added as Supplementary Figure 20 in the revised manuscript.

14. In the sentence “the energy barrier was obtained by subtracting the energy before the move from that after the move” of supplementary Note3, does the energy mean all energy in the system?

Reply and revision: The energy is the system energy, but the description was not correct in the original manuscript. We really appreciate the referee to point this out.

Accordingly, we changed the description in Supplementary Note 3 in the revised manuscript as follows:

“Each point in **b** is the system energy calculated, while the energy barrier can be obtained from the maximum energy value along the designated transport route.”

15. For Fig. S20, is there any quantitative correlation between the On/Off Switch Height and the hydrating ion radius of different ions?

Reply and revision: We really appreciate the referee's valuable comments. The referee raised a very interesting point that the On/Off switch height can be correlated to the hydrating ion radius of different ions. We have the same perspective as the referee suggested. However, we noticed that the error bar of On/Off switch height is relatively large, so that the quantitative correlation may not be as accurate as we expected. Therefore, this point has been qualitatively explained in our supplementary note 5 in the supporting information.

16. Please change the term "Van der Waals" to "van der Waals".

Reply and revision: We really appreciate the referee to point this out. We have changed the term "Van der Waals" to "van der Waals" in our revised manuscript.

17. According to the theoretical computations, the "ON" state shows the bulk movement of water molecules, which seems not the same as the movement of Newton's cradle-like Grotthus conduction. Authors should consider whether this description is appropriate or not.

Reply and revision: We really appreciate the referee's valuable comments. **Actually, we can see the bulk movement of water molecules in the "ON" state to be the continuous mode of Newton's cradle-like Grotthus Conduction.** The new Fig. 6b (previous Fig. 2c) represent two continuous states for the collision of one extra molecule onto the 2D ordered water nanosheets, leading to almost instant transport of the water molecule to the other side of the capillary. This is analogous to Newton's cradle-like Grotthus Conduction. More importantly, the two continuous states in Fig. 6b together represent only one snapshot for each collision. In real operations, it should be continuous collision of water molecules from the left, resulting in instant but continuous water transport to the other side of the capillary. Thus, the continuous mode of this analogy forms bulk flow of water.

Although the highly hydrogen-bonded network of water molecules formed within the atomic-scale graphene capillaries would not be as solid as those cradle balance balls, but we believe this analogy grasp the essence of the almost instant transport of water molecules from one side to the other side through the capillary.

Accordingly, we keep this analogy and add more descriptions on Page 19 of the revised manuscript to be,

"The bulk movement of water molecules in the "ON" state is then the outcome of the continuous mode of Newton's cradle-like Grotthus Conduction."

18. The shaded and solid sections of the bar chart in Fig 4c should indicate the meaning clearly.

Reply and revision: We really appreciate the referee's valuable comments. Fig. 5c (previous Fig. 4c) has been redrawn as shown in **Figure R15** for clear presentation.

Figure R15. Ultrafast water flux and effective ion sieving achieved simultaneously for various ions on MOGMs.

19. *How does the graphene capillaries size influence ion diffusion and water transport.*

Reply and revision: We greatly appreciate the referee's valuable comments. This is also a very interesting point raised by the referee. While it remains challenging to experimentally adjust the size of graphene capillaries beyond 0.6 nm at the atomic scale, we can explore the influence of graphene capillary size on ion diffusion and water transport *via* simulations. Fig. 4c shows the graphene capillaries with the *d*-spacing of 0.6 nm and Supplementary Figure 19 shows the graphene capillaries with the *d*-spacing of 0.87 nm. The former would form one layer of 2D crystal-like water nanosheet, while the latter would form two layers of 2D crystal-like water nanosheets.

In Supplementary Figure 22 and Supplementary Movies 3 and 4, we have further demonstrated the influence of the size of graphene capillaries on ion diffusion and water transport (see the Supplementary Note 4). Increasing the graphene capillaries further over the *d*-spacing of 1 nm, the structure of the included water molecules starts to transit from 2D crystal-like water nanosheet to bulk liquid water. In this case, the graphene nanoconfinement effect would vanish accordingly.

20. *The authors are suggested to check through the manuscript to improve the English. It is difficult to read.*

Reply and revision: We really appreciate the referee to point this out. We have reorganized the whole manuscript and improve the English in the revised version. Hope this version would be clear enough for readers to grasp the highlights and essence of our work easily.

21. Many recent publications related to ion transport behavior, such as Nat Commun 2022, 13, 6709, Nano Letters 2020, 20: 6937–6946, Joule 2023, 7: 251–253. In order to help readers better understand the importance of this work, these references should be cited.

Reply and revision: We really appreciate the referee's valuable comments. These references are indeed very important and we added them in our revised manuscript as references 15, 24, and 25.

Reviewer #3:

This manuscript describes a clever catalytic synthesis method to fabricate a sub-nano-confined, graphene-based microstructure. Ions can be selectively sieved by gating water molecules on/off. The mechanism is presented that water forms crystal-like water nanosheets inside the graphene capillaries and the application of a small hydrostatic pressure causes the water nanosheet to slide as a whole, enabling fast transport. This is very interesting. In addition, combing the experimental data with MD simulation has greatly improved the quality of this work. The manuscript can be considered for publications after minor revisions, as the comments listed below.

Reply and revision: We greatly appreciate the positive comments and all the very helpful suggestions from the reviewer.

1. How to make sure there is only one layer of Ni-pPD on top of rGO in assisting the formation of sub-nano-confined graphene capillaries?

Reply and revision: Thank you for the referee's valuable comment. The most significant evidence for this is BET and pore size distribution results shown in Fig. 2f and Supplementary Fig. 13 in the revised manuscript. The strongest peak in Fig. 2f sits at the pore size of 6.0 Å, although the pores smaller than 4.5 Å cannot be detected by CO₂ gas. The pores with the size of 6.0 Å can only be formed in the case that there is only one layer of Ni-pPD on top of rGO, which corresponds well with XRD, TEM and the simulation results in Figs 1 and 2.

Two other peaks below 1 nm in Fig. 2f imply that more than one layer of Ni-pPD may exist on top of rGO. However, because those peaks are relatively weak compared with the main peak at 6.0 Å, it is confident that most nano-islands on rGO nanosheets are monolayer Ni-pPD. Also, because the smallest pore size is the determining factor for molecular transport through the macroscopic-scale membranes, the discussion and conclusions made in the main text would not be affected by the pores with large sizes.

Figure R10. AFM analysis showing monolayer island structure on top of the spontaneously restacked Ni-pPD@rGO nanosheets. a, AFM image. b, Height profile. The fine monolayer island structure has been highlighted by pink circles.

To further confirm the monolayer of Ni-pPD on top of rGO, we also checked the fine structure of Ni-pPD monolayers on top of the restacked Ni-pPD@rGO nanosheets by AFM analysis. Please see the above figure (**Figure R10**), which is consistent with HAADF image in Fig. 1c in the revised manuscript.

Accordingly, the above figure has been added as Supplementary Figure 8 in the revised manuscript. The following descriptions have been revised in Supplementary Note 1 on Page 14 in the Supporting Information to be,

“The strongest peak sits at the pore size of 6.0 Å, corresponding well with XRD and the simulation results, although the pores smaller than 4.5 Å cannot be detected by CO₂ gas. Two other peaks below 1 nm in Fig. 2f may imply that some sub-nano graphene capillaries are formed with three or four layers of Ni-pPD nano-islands sandwiched between. However, because those peaks are relatively weak compared with the main peak at 6.0 Å, it is confident that most nano-islands on rGO nanosheets are monolayer Ni-pPD. Also, because the smallest pore size is the determining factor for molecular transport through the macroscopic-scale membranes, the discussion and conclusions made in the main text would not be affected by the pores with large sizes.”

2. *The process of ON/OFF is unclear. The authors should clearly define how the pressure is applied.*

Reply and revision: Thank you for the referee’s valuable comments. The hydrostatic pressure was applied by increasing the level (height) of the water on the feed (water side) compartment. Over the threshold pressure, the ultrafast water flux will be initiated.

Accordingly, the following description has been added on Page 12 of the revised manuscript,

“More importantly, water flow can be turned on and off reversibly (Fig. 3b), by adjusting hydrostatic pressures higher or lower than the gating pressures (P_G : $\sim 10^{-2}$ bar), revealing the successful establishment of a new gating mechanism for precise and ultrafast molecular sieving.”

3. *The membrane is not a single capillary, how to rationalize the simulation with only one graphene capillary without considering the tortuous pathways along ~2.3-micron thick graphene membranes?*

Reply and revision: We really appreciate the referee’s valuable comments. It is true that the ~2.3-micron thick graphene membranes have large tortuosity where the water flow will also be influenced by the edge carbon atoms, dangling bonds and hydrogen bonding between the functional groups; **however, the key structure to generate frictionless water flow in MOGM membranes is the atomic-scale graphene capillaries.** As discussed in our manuscript, the edge carbon atoms, dangling bonds and hydrogen bonding between the functional groups **would not induce the frictionless flow of water molecules but provide resistance to the water flow.**

This resistance can be attributed to the gating pressures (or energy barriers) that we need to overcome in initiating ultrafast water flux (Supplementary Fig. 25).

Accordingly, the following descriptions have been revised on Pages 19 and 20 in revised manuscript to be,

“Therefore, although the micrometer-thick MOGMs have long and torturous permeation paths, it is this high-speed, Grotthus conduction that enables ultrafast water flux.”

“In addition to the edge carbon atoms, dangling bonds, and the interstitial water/ions outside of the graphene capillaries that could add resistance to the bulk water flow, Supplementary Fig. 23 also implies that the counter motion of ions, although suppressed, also exert resistance to the sliding of the ordered water nanosheets. All these factors can contribute to the magnitude of gating pressures (Supplementary Fig. 25).”

4. *The resolution in several figures is not high enough. For example, Fig. 1b.*

Reply and revision: Thank you for the referee’s valuable comments. We tried many advanced characterization techniques for several times, but still, it is very challenging to obtain the atomic resolution of the heterostructure with the following reasons. First, in STEM or HAADF, our samples are moving under the electron beam with high accelerating voltages. Second, the monolayer of Ni-pPD itself is in amorphous structure. Third, C atoms are light elements and there is only one monolayer of Ni-pPD on top of rGO. The signals from these two layers would interact with each other, preventing us from clear demonstration of the atomic resolution of the heterostructure.

Although the atomic resolution of the heterostructure is difficult to obtain, the main features of the heterostructure can be determined clearly. In addition to HRTEM and HAADF characterization, we also detected distinct features from XRD and BET/Pore size distribution, all suggesting that the heterostructure is composed of Ni-pPD monolayer-islands distributed uniformly on rGO nanosheets. DFT calculations were further conducted to understand the microstructures and how the heterostructure formed, which correspond well with the experimental results.

5. *How is the comparison between the DFT simulation results and experimental results?*

Reply and revision: We really appreciate the referee’s valuable comments. DFT simulation results present in our work agree well with our experimental results.

First, the optimized structure computed by DFT (Supplementary Fig. 9) demonstrates that two layers of Ni-pPD are pillared between two rGO through π - π stacking with all the stacking distances being in the range from 0.34 to 0.36 nm, generating the interlayer spacing of 0.96 nm from the top to the bottom. This result agrees well with the XRD result in Fig. 2e and HRTEM result in Fig. 2c in the revised manuscript.

Second, as shown in the simulated results in Fig. 2d, the d -spacing between the 1st-layer rGO and 3rd-layer pPD was calculated to be 6.0 Å, giving the free space of 2.6 Å (marked as pink slash lines in Fig. 2d); and also, the d -spacing between the 1st-layer rGO and 4th-layer rGO was determined to be 9.6 Å, giving the free space of 6.2 Å (marked as blue slash lines in Fig. 2d). This result corresponds well with the BET/Pore size distribution result, since Fig. 2f clearly demonstrates three sharp and distinctive peaks below 1 nm, with the strongest peak sitting at ~0.6 nm.

Third, the optimized arrangements of water molecules within 6.0 Å-interspaced graphene capillary in Fig. 4c show a monolayer of 2D ordered structure. The average number of hydrogen bonds around each water molecule has been calculated to be $\text{HB}\cdot\text{H}_2\text{O} = 4.83$ in Supplementary Fig. 20. The in-situ FTIR shows the increased intensity of blue and cyan peaks for MOGMs in 37% humidity, which correspond to strongly-bonded water network included with $\text{HB}\cdot\text{H}_2\text{O} > 4$. Thus, DFT simulation results are also consistent with the experimental results.

Based on these, all molecular dynamics (MD) simulations conducted in this work reflect the real case of our observations in the permeation test, which gives us deep understanding of the anomalous water molecular gating mechanism. Therefore, all our DFT simulation results agree well with our experimental results.

6. The flux data on Figure S13 extrapolate to zero LMH at zero applied hydrostatic pressure, while for the data on the Figure S12, the data extrapolate to about 4LMH. What is the reason for such a difference?

Reply and revision: Thank you for the referee's valuable comments. We think the referee raised a very interesting point that we did not pay attention to. When extrapolating the water flux data to zero applied hydrostatic pressure, Supplementary Fig. 16 (previous Supplementary Fig. 13) shows zero LMH for 0.1 M KCl draw solution, while Supplementary Fig. 15 (previous Supplementary Fig. 12) shows 4 LMH for 0.1 M NaCl draw solution. This observation means that the slope of the water flux data in the "ON" state is much smaller for the case of 0.1 M NaCl than that for the case of 0.1 M KCl. **This is actually another experimental proof that the resistance for the sliding of the 2D water nanosheet to the draw (salt side) is dependent on the type of the draw solutions.** In other words, the resistance is higher for the draw solution of 0.1 M NaCl than that for the draw solution of 0.1 M KCl. We really appreciate the referee's thoughtful considerations.

Accordingly, the following descriptions have been added into Supplementary Note 5 on Page 24 in Supporting Information to be,

"When extrapolating the water flux data to zero applied hydrostatic pressure, Supplementary Fig. 16 shows zero LMH for 0.1 M KCl draw solution, while Supplementary Fig. 15 shows 4 LMH for 0.1 M NaCl draw solution. This observation means that the slope of the water flux data in the "ON" state is smaller for the case of 0.1 M NaCl than that for the case of 0.1 M KCl. This is also

another experimental proof that the resistance for the sliding of the 2D water nanosheet to the draw (salt side) is dependent on the type of the draw solutions.”

7. *The authors refer to the nanosheets as graphene at times which is not the correct use of terminology. The authors should choose and use the appropriate term.*

Reply and revision: We greatly appreciate the valuable comments. The terminology has been corrected to be “nanosheets or rGO” in the revised manuscript. The term “graphene” has only been used to describe the capillaries, which are atomic-scale graphene capillaries.

8. *There are several errors in references formats.*

Reply and revision: We really appreciate the referee’s valuable comments. We have corrected the errors of the references’ formats in the revised manuscript. We would like to thank the referee again for the careful review of these typos.

References:

- [R1] Liu, X. et al. 2D material nanofiltration membranes: from fundamental understandings to rational design. *Adv. Sci.* **8**, 2102493 (2021).
- [R2] Wilson, N. R. et al. Graphene oxide: structural analysis and application as a highly transparent support for electron microscopy. *ACS Nano* **3**, 2547–2556 (2009).
- [R3] Radha, B. et al. Molecular transport through capillaries made with atomic-scale precision. *Nature* **538**, 222–225 (2016).
- [R4] Liu, H. Y., Wang, H. T. & Zhang, X. W. Facile fabrication of freestanding ultrathin reduced graphene oxide membranes for water purification. *Adv. Mater.* **27**, 249–254 (2015).
- [R5] Zheng, S. X., Tu, Q. S., Urban, J. J., Li, S. F. & Mi, B. X. Swelling of graphene oxide membranes in aqueous solution: characterization of interlayer spacing and insight into water transport mechanisms. *ACS Nano* **11**, 6440–6450 (2017).
- [R6] Liu, J., He, X., Zhang, J. Z. H. & Qi, L. W. Hydrogen-bond structure dynamics in bulk water: insights from *ab initio* simulations with coupled cluster theory. *Chem. Sci.* **9**, 2065–2073 (2018).

: ** If you wish to forward this email to your co-authors, please delete the link to your author home page below **

Dear Professor Zeng,

Thank you again for submitting your revised manuscript "Anomalous Water Molecular Gating from Atomic-scale Graphene Capillaries for Precise and Ultrafast Molecular Sieving" to Nature Communications. We have now received reports from the three reviewers who evaluated the original version. On the basis of their comments (copied below), we have decided to invite an additional revision of your work.

You will see that, while the reviewers find that your revisions improved the manuscript, some important points remain to be addressed. In particular, any discrepancies between values reported in the manuscript should be addressed. Please revise your manuscript, addressing all the remaining issues raised by the reviewers.

HOW TO SUBMIT

Please use the link below to submit the following items as separate documents:

- Revised manuscript
- Any supplementary files
- Point-by-point response to the reviewers' comments, reproduced verbatim
- Cover letter to the editor
- Updated editorial policy checklist (see below for details)
- Updated reporting summary (see below for details)
- Updated source data file (see below for details)

<*** DELETE REPORTING SUMMARY/SOURCE DATA REQUEST OR MODIFY AS REQUIRED ***>

[https://mts-ncomms.nature.com/cgi-](https://mts-ncomms.nature.com/cgi-bin/main.plex?el=A1S2DBdu1B6DSqh4I2A9ftds5emT1OX8fhMqFnBxKlpwZ)

[bin/main.plex?el=A1S2DBdu1B6DSqh4I2A9ftds5emT1OX8fhMqFnBxKlpwZ](https://mts-ncomms.nature.com/cgi-bin/main.plex?el=A1S2DBdu1B6DSqh4I2A9ftds5emT1OX8fhMqFnBxKlpwZ)

** If you wish to forward this email to your coauthors, please delete the link above **

We hope to receive your revised paper within four weeks, but we understand that revisions may take longer. Please let us know if you find that the revision process will take substantially more time.

Best regards,

Harry

Dr Harry Geddes
Associate Editor
Nature Communications
The Campus, 4 Crinan Street, London N1 9XW, UK

POLICIES AND FORMS REQUIRED FOR RESUBMISSION

* Please update the following checklist(s) to verify compliance with our research ethics and data reporting standards. Address all points on the checklist, revising your manuscript in response to the points if needed.

The form(s) must be downloaded and completed in Adobe Reader rather than opened in a web browser. Each form must be uploaded as a Related Manuscript file at the time of resubmission.

Editorial policy checklist:

<https://www.nature.com/documents/nr-editorial-policy-checklist.pdf>

DATA AND CODE AVAILABILITY

* All Nature Communications manuscripts must include a "Data Availability" section after the Methods section but before the References. If any of the data can only be shared on request or are subject to restrictions, please specify the reasons and explain how, when, and by whom the data can be accessed. For more information on this policy and a list of examples, see: <https://www.nature.com/documents/nr-data-availability-statements-data-citations.pdf>

* As Nature Portfolio [policies](https://www.nature.com/nature-portfolio/editorial-policies/reporting-standards#availability-of-data) strongly encourage you to share your research data in a public repository (e.g. spreadsheets, text, images), we are partnering with the figshare repository so that you can use the figshare integration via the 'Research Data Deposition' tab when submitting your revised manuscript.

Data are stored privately until a manuscript decision is reached and you can edit/withdraw them up to this point: you retain rights and control over your data. The data will be published at the same time as your article; you will receive a data DOI, with guidance on linking the data and manuscript. In the event your manuscript is not accepted, you can keep or remove your data in figshare.

We recommend the use of discipline-specific repositories where available and for a number of [data types](https://www.nature.com/nature-portfolio/editorial-policies/reporting-standards#availability-of-data) this is mandatory. Ensure you do not submit these data types or any sensitive data to figshare.

* We strongly encourage you to deposit all new data associated with the paper in a persistent repository where they can be freely and enduringly accessed. We recommend submitting the data to discipline-specific and community-recognised repositories; a list of repositories is provided here: <http://www.nature.com/sdata/policies/repositories>
Refer to our data policies here: <https://www.nature.com/nature-portfolio/editorial-policies/reporting-standards#availability-of-data>

* To maximise the reproducibility of research data, we strongly encourage you to provide a file containing the raw data underlying the following types of display items:

- Any reported means/averages in box plots, bar charts, and tables
- Dot plots/scatter plots, especially when there are overlapping points
- Line graphs

The data should be provided in a single Excel file with data for each figure/table in a separate sheet, or in multiple labelled files within a zipped folder. Name this file or folder 'Source Data', and include a brief description in your cover letter. The "Data Availability" section should also include the statement "Source data are provided with this paper."

To learn more about our motivation behind this policy, please see:
<https://www.nature.com/articles/s41467-018-06012-8>

ORCID

* Nature Communications is committed to improving transparency in authorship. As part of our efforts in this direction, we are now requesting that all authors identified as 'corresponding author' create and link their Open Researcher and Contributor Identifier (ORCID) with their account on the Manuscript Tracking System prior to acceptance. ORCID helps the scientific community achieve unambiguous attribution of all scholarly contributions.

You can create and link your ORCID from the home page of the Manuscript Tracking System by clicking on 'Modify my Springer Nature account' and following [these instructions](https://www.springernature.com/gp/researchers/orcid/orcid-for-nature-research). Please also inform all co-authors that they can add their ORCIDs to their accounts and that they must do so prior to acceptance.

REVIEWER COMMENTS

Reviewer #1 (Remarks to the Author):

Comments attached in PDF

Reviewer #2 (Remarks to the Author):

The authors have carefully addressed all the issues I raised previously. I recommend it for publication.

Reviewer #3 (Remarks to the Author):

I am satisfied with the revision and can accept the manuscript for publication in its current form.

RESPONSE TO REVIEWERS' COMMENTS

Reviewer #1:

Authors have revised the manuscript significantly, especially to address the synthesis and characterization of the membranes. They have addressed most of the queries, however the following remain to be answered. I would suggest them to address the below before their manuscript can be accepted.

Reply and revision: We greatly appreciate the valuable comments and all the very helpful suggestions from the reviewer.

1. In the abstract, it is mentioned, “.....graphene capillaries at 6 Å, which were.....” However, in Figure 2d and Supplementary Figure 9, it is mentioned as 6.2 Å. In view of this, the authors can put an error and describe it. Perhaps, does the Ni-pPD@rGO membrane expand in water?

Reply and revision: Thank you for the referee's valuable comments. Graphene capillaries with the size of ~6 Å were determined experimentally through pore size distribution (PSD) tests of MOGMs as shown in Figure 2f. However, the size of 6.2 Å was calculated from DFT analyses. In experiment, it is very hard to detect 0.2 Å difference in pore size, and thus, we claim that the result from DFT calculations corresponds well with our experimental investigations. Furthermore, this little difference has no relation to the membrane expansion in water.

However, we agree with the referee that this 0.2 Å difference may cause confusions to the readers. Accordingly, we added the following comment on Page 10 of the Revised Manuscript as follows:

“Fig. 2f clearly demonstrates three sharp and distinctive peaks below 1 nm, with the strongest peak sitting at ~0.6 nm. Considering the calculated free space of 6.2 Å in Fig. 2d, the PSD experimental results in Fig. 2f correspond well with our DFT calculations.”

2. Regarding the zeta potential of the GO and the Ni-pPD@rGO, please describe in brief how they are measured. For instance, what was the concentration of the nanosheets dispersion. Also, please include the zeta potential of the Ni-pPD complex in Supplementary Fig. 6.

Reply and revision: Thanks for the referee's critical comments. To investigate the change of surface charge before and after the designated reaction, the GO and the Ni-pPD@rGO were dispersed in Milli-Q water, respectively, forming the colloidal solutions at the concentration of 0.05 mg/ml for zeta potential tests.

Accordingly, the detailed experimental procedures of zeta potential tests have been added in the experimental section on Page 23 of the Revised Manuscript as follows:

“Zeta potential was performed on Malvern zeta sizer nano series, with Ni-pPD@rGO nanosheets, GO nanosheets, and pure Ni-pPD particles being dispersed in Milli-Q water in the concentration of 0.05 mg/ml.”

As also suggested by the referee, the zeta potential of the Ni-pPD complex has been detected using the same procedure. This result has been added into Supplementary Fig. 6 on **Page 7 of Supporting Information in the Revised Manuscript**, which is also shown as Figure R1 below:

Figure R1. Zeta potential analyses of Ni-pPD@rGO nanosheets in comparison with GO nanosheets and pure Ni-pPD particles.

3. In response to the previous question in review about the free space, “As claimed in Figure 1h, the d -spacing of the Ni-pPD@rGO is 9.6 \AA , which gives a free space of $\sim 6.2 \text{ \AA}$ (subtracting the basal plan thickness 3.4 \AA from d -spacing). Now, it is not clear how two Ni-pPD sheets each of 2.6 \AA will fit between two GO flakes when the space between a GO and the Ni-pPD is 3.63 \AA (Figure 1d). Even neglecting the space between two Ni-pPD monolayers, the total free space between two GO layers should be 12.46 \AA .”

The authors have replied “Fig. 1e (previous Fig. 1d) may be a little misleading. 3.63 \AA in Fig. 1e is actually the d -spacing between the rGO and the restacked Ni-pPD complex, not the free space. This is understandable, because it is the normal π - π stacking. Therefore, three layers’ π - π stacking, including rGO-pPD, pPD-pPD, and pPD-rGO, should result in the d -spacing of the restacked Ni-pPD@rGO being around $3 \times 3.63 = 10.9 \text{ \AA}$ (Figure R9a). Furthermore, as also shown in Figure R9a, because of small interaction between the 1st-layer rGO and 3rd-layer pPD, rGO and pPD nanosheets will be deformed a little bit, contracting the whole d -spacing to be 9.6 \AA . In this way, the free space between the 1st-layer rGO and 3rd-layer pPD is $6.01 \text{ \AA} - 3.4 \text{ \AA} = 2.61 \text{ \AA}$, and that between the 1st-layer rGO and 4th-layer rGO is $9.58 \text{ \AA} - 3.4 \text{ \AA} = 6.18 \text{ \AA}$ (Figure R9b).”

However, it is not clear but is more confusing. If there were two Ni-pPD islands in between the rGO, there would be almost no space (about 1 angstrom) as seen in supplementary Figure 9. In fact with this calculation, one can say that graphene capillaries are conducting only along the regions devoid of the Ni-pDP as illustrated in the revised figure 2g. The top view schematic is

clearer however the cross-sectional schematics do not convey the correct spacings, especially as the numbers do not add up (Supplementary Figure 9, and figure 2d). In principle rGO should not have free space enough for water transport, however with their Ni-pDP pillars the authors have sufficiently maintained the rGO distance to allow water transport. This message is not conveyed clearly and is lost in the details.

In the revised manuscript on page 8, line 148 onwards, the authors wrote “Considering a monolayer of Ni-pPD nano-islands formed on each side of rGO, the repeating unit of the restacked structure between two rGO should be around 1 nm”. How is this 1 nm gotten?

If there are two monolayers of Ni-pPD between two rGO sheets), **the channels should be blocked at the interface of the two monolayers of Ni-pPD** (as shown in Fig 2d and Supplementary Figure 9). This is not clearly conveyed.

There is a lack of proper explanation, which needs to be rectified. A proper schematic is required to show the nanofluidic channels between two rGO sheets (like in Figure 2g). The present schematic is confusing as it is showing the channels are blocked. What is the pathway of the ion transport, along the plane or perpendicular to the plane of the rGO?

Reply and revision: Thank you for the valuable comments. We agree with the referee that “one can say that graphene capillaries are conducting only along the regions devoid of the Ni-pDP”. Those graphene capillaries are the ones with the size of $\sim 6.0 \text{ \AA}$ (Figure R2a).

The referee mentioned that “If there were two Ni-pPD islands in between the rGO, there would be almost no space”. This would be the case when two Ni-pPD-island nanolayers match their size and position exactly between two rGO nanosheets (Figure R2a). However, when the two Ni-pPD-island nanolayers do not match their size and position (Figure R2b), there would be another type of graphene capillaries created at around 2.6 \AA , according to our DFT calculations about the cross-sectional view of the restacked structure in Figure 2d.

Figure R2. Cross-sectional illustration of two kinds of the restacking situations with two monolayers of Ni-pPD nano-islands between two rGO nanosheets.

Therefore, in the top view schematic Figure 2g, there are three kinds of regions between two rGO nanosheets. One type contains two monolayers of Ni-pPD, which, as also mentioned by the referee, blocks the channel completely. The second type is the region free of any Ni-pPD islands, resulting in the graphene capillaries at $\sim 6 \text{ \AA}$. There is a third type of region, where only one monolayer of Ni-pPD exists within two rGO nanosheets, resulting in graphene capillaries at 2.6 \AA for molecular transport.

According to the referee's suggestion, the top view schematic in Figure 2g (as also shown in Figure R3) was redrawn to show the fully-pillared (blocked) region by pink-shaded frames, leaving 2.6 Å and 6.0 Å graphene capillaries for molecules to transport. Therefore, the pathway of the water/ion transport is along the plane of the rGO, labelled by the red arrows.

Figure R3. Schematic of the sub-nano pathways to conduct water molecules in MOGMs. Green: bottom Ni-pPD nano-islands; Yellow: top Ni-pPD nano-islands; Pink-shaded frames: fully blocked regions by the restacking of two Ni-pPD nano-islands.

The case that the numbers do not add up in Supplementary Figure 8 (previous Supplementary Figure 9) and Figure 2d is due to the distortion of Ni-pPD monolayers within two rGO nanosheets, which is clearly presented in Supplementary Figure 8a (also in Figure R4). This phenomenon is also caused by the mismatch of the two Ni-pPD monolayers, in which case the 4th layer rGO will attract the 2nd layer Ni-pPD to some extent.

Figure R4. Cross-sectional illustration (a) and DFT calculated result (b), showing the distortion of the Ni-pPD monolayer caused by the attraction between the 2nd layer Ni-pPD and the 4th layer rGO.

We also agree with the referee that “In principle rGO should not have free space enough for water transport, however with their Ni-pDP pillars the authors have sufficiently maintained the rGO distance to allow water transport. This message is not conveyed clearly and is lost in the details.” Accordingly, we added this sentence on Page 8 in the revised manuscript to emphasize the key message we tried to deliver in this work.

“Therefore, although the restacked rGOs should not present free space, the rGOs pillared by Ni-pDP nano-islands can sufficiently maintain their distance for molecular transport.”

Since Ni-pPD is mainly composed of four benzene rings in a plane on top of rGO, the theoretical molecular size of the Ni-pPD nano-islands is similar to that of a monolayer of graphene, 3.4~3.6 Å. In this case, there are three layers of π - π stacking for the repeating unit of the restacked structure between two rGO. They are, 1) the 1st layer rGO with benzene rings in the 2nd layer Ni-pPD, 2) benzene rings in the 2nd layer Ni-pPD with benzene rings in the 3rd layer Ni-pPD, 3) benzene rings in the 3rd layer Ni-pPD with the 4th layer rGO, resulting in $0.34\sim 0.363\text{ nm} \times 3 = 1.02\sim 1.09\text{ nm}$.

Why the 002 peak of the MOGM in Figure 2e is very wide? Does it indicate the presence of rGO-pPD-rGO along with rGO-pPD-pPD-rGO orientation?

We totally agree with the referee’s understanding. It is the case of “rGO-(Ni-pPD)-rGO” that causes the distortion of Ni-pPD monolayers within two rGO nanosheets, leading to the numbers that do not add up in Supplementary Figure 8 and the graphene capillaries at 2.6 Å in Figure 2d.

4. “The pore size distribution (PSD) analysis was conducted using the slit pore geometry” How the sample were prepared for this experiment? Is it in MOGMs form? Please comment on the pore size of 0.87 nm and 1 nm, shown in Fig 2f.

Reply and revision: The referee is right. The samples prepared for pore size distribution (PSD) tests are in their freestanding MOGMs form, in order to detect the graphene capillaries formed after the restacking of Ni-pPD@rGO nanosheets.

Accordingly, the detailed experimental procedures of PSD tests have been added in the experimental section **on Page 23 of the Revised Manuscript** as follows:

“Gas-adsorption measurements were made using Micromeritics ASAP 2020 using N₂ gas at 77K as well as carbon dioxide at 273.2 K. Before the gas-adsorption experiment, freestanding MOGMs were thoroughly washed and dried, and then “outgassed” under vacuum at 150 °C for 8 h.”

In addition, according to the referee’s valuable suggestion, the comments regarding the pore size of 0.87 nm and 1 nm shown in Figure 2f have been added **on Page 13 of Supporting Information in the Revised Manuscript**, as follows:

“**Supplementary Note 1:** The strongest peak sits at the pore size of 6.0 Å, corresponding well with XRD and the simulation results, although the pores smaller than 4.5 Å cannot be detected by CO₂ gas. Two other peaks below 1 nm in Fig. 2f may imply that some sub-nano graphene capillaries are formed with three or four layers of Ni-pPD nano-islands sandwiched between two

rGO nanosheets. However, because those peaks are relatively weak compared with the main peak at 6.0 Å, it is confident that most nano-islands formed on rGO nanosheets are monolayer Ni-pPD. Also, because the smallest pore size is the determining factor for molecular transport through the macroscopic-scale membranes, the discussion and conclusions made in the main text would not be affected by the pores with larger sub-nano sizes. In addition, the pore size at around 2.5 nm should be considered as the interedge pores between two separated Ni-pPD@rGO nanosheets aligned side by side^[R1].”

5. AFM image in figure 1 – the surface is very smooth whereas with similar height of the flakes supplementary figure 8 is claimed to be rough and that the roughness is due to the monolayer of Ni-pPD on rGO. What is the theoretical molecular size of the Ni-pPD? Does it match with the authors’ claim, “a monolayer of Ni-pPD nano-islands formed on each side of rGO, the repeating unit of the restacked structure between two rGO should be around 1 nm?”

It may be too much of a stretch in analysis to correlate the roughness on some areas of the rGO flakes to be due to the Ni-pPD whereas other areas of the flakes which are equally rough are not considered to be coated with Ni-pPD. For instance, why is the roughness only in the circled area (supplementary Figure 8 reproduced below for convenience) is considered to be coated with Ni-pPD as all through the flake is equally rough.

Reply and revision: Thank you for the referee’s critical comments. Since Ni-pPD is mainly composed of four benzene rings in a plane on top of rGO, the theoretical molecular size of the Ni-pPD nano-islands is similar to that of a monolayer of graphene, 3.4~3.6 Å. This is also confirmed by our DFT calculations in Figure 1e, and by their restacked structures in Figure 2d and Supplementary Figure 8 (previous Supplementary Figure 9).

There are three layers of π - π stacking for the repeating unit of the restacked structure between two rGO. They are, 1) the 1st layer rGO with benzene rings in the 2nd layer Ni-pPD, 2) benzene rings in the 2nd layer Ni-pPD with benzene rings in the 3rd layer Ni-pPD, 3) benzene rings in the 3rd layer Ni-pPD with the 4th layer rGO, resulting in $0.34\sim 0.363\text{ nm} \times 3 = 1.02\sim 1.09\text{ nm}$. Therefore, we claim that “a monolayer of Ni-pPD nano-islands formed on each side of rGO, the repeating unit of the restacked structure between two rGO should be around 1 nm”, and it matches the theoretical molecular size of the Ni-pPD nano-islands.

We do agree with the referee that it may be too much of a stretch in AFM analyses, and thus, we deleted the related discussion and supplementary figure in the revised manuscript.

Thanks to the referee’s valuable suggestion.

Reviewer #2:

The authors have carefully addressed all the issues I raised previously. I recommend it for publication.

Reply and revision: We greatly appreciate the positive comments and all the very helpful suggestions from the reviewer.

Reviewer #3:

I am satisfied with the revision and can accept the manuscript for publication in its current form.

Reply and revision: We greatly appreciate the positive comments and all the very helpful suggestions from the reviewer.

References:

[R1] Liu, X. et al. 2D material nanofiltration membranes: from fundamental understandings to rational design. *Adv. Sci.* **8**, 2102493 (2021).

REVIEWERS' COMMENTS

Reviewer #1 (Remarks to the Author):

Authors have answered all the queries.

RESPONSE TO REVIEWERS' COMMENTS

Reviewer #1:

The manuscript entitled “Anomalous Liquid Gating from Atomic-scale Graphene Capillaries for Precise and Ultrafast Molecular Sieving” has many issues which need to be rectified before submitting it to any scientific journal. At the current stage, it is not suitable for publication in Nature Communication. Some (not all) of the major issues are listed below.

Reply and revision: We greatly appreciate the valuable comments and all the very helpful suggestions from the reviewer.

1. The claim “for the first time, allows us to take full advantage of atomic-scale, graphene capillaries under osmosis at room temperature” is misleading. Firstly, the manuscript is on reduced graphene oxide capillaries. Secondly, one of the authors Prof Lei Jiang group are already pioneer in the power generation from GO membranes, including osmosis. Hence the claim is inappropriate.

Reply and revision: Thank you for the referee’s valuable comments. Both oxidized graphene capillaries and pristine graphene capillaries are present in GO membranes (GOMs) and rGO-based membranes, as shown in **Figure R1a**. It is noted that **there are no capillaries specifically designated as reduced graphene oxide (rGO) capillaries; instead, rGO-based membranes contain a higher number of pristine graphene capillaries than those in GOMs.** [R¹]

Figure R1. (a) Illustration of oxidized graphene capillaries and pristine graphene capillaries in GOMs and rGO-based membranes. (b) Reduced graphene region tends to restack, sealing the atomic-scale, graphene capillaries.

Moreover, although pristine graphene capillaries are present in both types of membranes, **the advantages of atomic-scale, graphene capillaries have not been fully explored under osmosis at room temperature.** As mentioned by the referee, GOMs have been tested for water-ion separation under osmosis. Nevertheless, GOMs exhibit a high hydrophilicity due to a significant presence of oxygenated functional groups (40-50% of the oxidized region). [R²] This abundance of functional groups can hinder the rate of water transport, thereby obscuring the

beneficial effects of atomic-scale graphene capillaries under osmosis at room temperature. On the other hand, rGO-based membranes contain significantly fewer oxygenated functional groups as compared to GOMs, resulting in a higher proportion of pristine graphene capillaries. However, the challenge for rGO-based membranes is the tendency of the reduced region to restack, effectively sealing the atomic-scale, graphene capillaries (**Figure R1b**), which restricts permeation only through large graphitic pores and deteriorates the ion sieving properties. Despite demonstrating fast frictionless flow of water, rGO-based membranes do not exhibit as efficient ion sieving as GOMs. **Therefore, the main challenge lies in the control of 2D graphene slit pore size with angstrom precision in rGO-based membranes.** In this work, we have successfully achieved this goal, resulting in the simultaneous achievement of ultrafast bulk water flow of 2.5-45.4 L m⁻² h⁻¹ and negligible ion permeation rates of ~10⁻⁴ mol m⁻² h⁻¹ for various alkali-metal ions. These remarkable properties are governed by a newly-discovered, anomalous water molecular gating mechanism.

Therefore, as suggested by the referee and to provide greater precision, we have revised the relevant description in our revised manuscript: “**For the first time, it enables us to fully exploit the potential of atomic-scale graphene capillaries in rGO-based membranes under osmosis at room temperature.**” In addition, we would like to clarify that our membranes are rGO-based membranes rather than GO-based membranes. The most direct evidence supporting this is the contact angle results shown in Fig. 2e. The contact angle of our membranes is 84°, which is very close to that of pure graphite (85°).^[R3] In contrast, GO-based membranes are hydrophilic with a contact angle of approximately 44.5°. Further details regarding this distinction have been included in our responses to the referee’s next two questions.

Accordingly, we revised the term “graphene-based nanofiltration (NF) membranes” to “rGO-based NF membranes” in the revised manuscript. In addition, we revised the relevant description on Page 4 of the Revised Manuscript as follows:

“For the first time, it enables us to fully exploit the potential of atomic-scale graphene capillaries in rGO-based membranes under osmosis at room temperature.”

2. It is quite surprising to know that the oxygenated functional groups disappear in the resulting nanosheets. GO nanosheets are known to have different oxygen-containing functional groups like –O–, –OH, –COOH, C=O, etc. It is very hard to reduce all these functional groups especially the –O– and –COOH even in harsh conditions. In the FTIR compared in the supplementary Figure 3, the authors have indicated the presence of the OH and the C-OH groups in the Ni-pPD@rGO nanosheets, which directly contradicts their claim of “the oxygenated functional groups disappear in the resulted nanosheets, implying the reduction of GO to rGO after the reaction.” Careful observation in the spectra also indicate the presence of other oxygen functional groups like C-O-C in the Ni-pPD@rGO nanosheets. In addition, the XPS (supplementary Figure 4a) spectra of the Ni-pPD@rGO nanosheets have a very strong

peak for O 1s at ~530 eV. This clearly indicates the presence of different oxygen functional groups.

Reply and revision: Thanks for the referee’s critical comments. We acknowledge the referee's interpretation and agree that Supplementary Figs 3 and 4a provide clear evidence of the presence of distinct oxygen functional groups on our Ni-pPD@rGO nanosheets. This is understandable because even after the reduction of GO to rGO, a small amount of oxygenated functional groups may still remain, as observed in other GO reduction methods.^[R4] Thus, the term “disappear” could potentially confuse the referee and potential readers. Therefore, we have revised the manuscript to clarify that “the oxygenated functional groups **have been substantially removed in the resulted nanosheets**, implying the reduction of GO to rGO after the reaction.”

Although different oxygen functional groups are present, the quantity of oxygenated functional groups in our Ni-pPD@rGO nanosheets has substantially decreased, compared to the original GO nanosheets. This reduction has also been confirmed through both FTIR and XPS analyses, as shown in Supplementary Figs 3, 4a, and 4b in the revised manuscript. Notably, in Supplementary Fig. 4b (also depicted in **Figure R2a**), the C 1s XPS spectrum of GO is compared with that of Ni-pPD@rGO nanosheets, revealing a substantial suppression of the oxygenated carbon peak at approximately 286.8 eV after the proposed reaction. This clear reduction in intensity indicates the removal of oxygenated functional groups from carbon to a significant degree. Moreover, the contact angle results depicted in Fig. 2e of the revised manuscript provide direct evidence of GO reduction. The contact angle of Ni-pPD@rGO membranes is 84°, which is very close to that of pure graphite (85°).^[R3] In contrast, GO-based membranes are hydrophilic with a contact angle of about 44.5° (**Figure R2b**).

Figure R2. (a) XPS analyses of Ni-pPD@rGO nanosheets in comparison with GO nanosheets. (b) Contact angle and XRD analyses. (c) Pore size distribution analysis.

More importantly, the presence of a small amount of oxygenated functional groups on rGO does not significantly impact the discussion and conclusions presented in our manuscript. **The reduction of GO to rGO does not imply that all capillaries become reduced graphene oxide capillaries. Rather, it signifies a substantial decrease in the percentage of oxidized graphene**

capillaries, resulting in a larger proportion of exposed pristine graphene regions. However, at the atomic scale, these pristine graphene capillaries tend to become sealed because of restacking phenomenon (**Figure R1b**), as reported in previous studies.^[R4] To address this challenge, we developed a novel method to form a Ni-pPD island-on-rGO nanosheet microstructure. The spontaneous restacking of Ni-pPD@rGO nanosheets leads to the creation of stable atomic-scale pristine graphene capillaries, predominantly with a size of 0.6 nm. This result has also been experimentally confirmed through pore size distribution analyses, as shown in Fig. 2f (**Figure R2c**). Therefore, the sub-nano graphene pore structure in our membranes exhibits similar characteristics to those of 2D molecular sieves, demonstrating the delicacy of this synthesis strategy and supporting our discussion regarding the anomalous water molecular gating mechanism.

Accordingly, we changed our claim to be “the oxygenated functional groups have been substantially removed in the resulted nanosheets, implying the reduction of GO to rGO after the reaction.” on **Page 6 of the revised manuscript.**

3. How did the authors confirm the formation of the Ni-pPD complex? They did not discuss it properly in any of the characterization techniques. For example, in the FTIR, which peak confirms the Ni-N bond (N=Q=N peak?). In XPS, the peak under O 1s at ~530 eV is not well described.

In all the characterization techniques, the authors compared the spectra of the GO and Ni-pPD@rGO nanosheets. Although the formation of the Ni-pPD complex is a known reaction (as mentioned in the ref 26), but this formation might be different in the presence of the GO nanosheets. For example, the presence of the solder peak in the Ni 2p spectra (supplementary Figure 4c) at ~862 eV clearly indicates the presence of the Ni(OH)₂ (ref 26).

Reply and revision: Thank you for the valuable comments. The Ni-pPD complex possesses distinct functional groups, including -C-N groups, -C=N groups, N=Q=N groups (Q representing the quinonoid ring), -NH-, -N=, and Ni-N. As also raised by the referee, N=Q=N groups are a particular substructure in Ni-pPD complex formed through the designated catalytic reaction, according to ref. 29 (previous ref. 26).

In order to confirm the formation of the Ni-pPD complex, we have conducted EDX mapping, FTIR and XPS analyses in this work. Firstly, STEM-EDX mapping images in Fig. 1b and Supplementary Fig. 2 show homogeneous distribution of N on the resulted nanosheets. Secondly, the characteristic peaks in FTIR Spectra (Supplementary Fig. 3) at 1274 cm⁻¹ and 1168 cm⁻¹ correspond to -C-N and N=Q=N groups, respectively. Furthermore, XPS analyses (Supplementary Fig. 4) also show the characteristic peaks of Ni-pPD (-C-N and -C=N groups in C 1s, -NH-, -N=, and Ni-N bonds in N 1s), consistent with the observations found in the FTIR results. **In particular, the presence of N=Q=N groups in the FTIR spectra and Ni-N bonds in the XPS spectra provides strong evidence for the formation of the Ni-pPD complex.**

The comparison of XPS and FTIR spectra of pure GO, pure Ni-pPD metal complex, GO treated with Ni(NO₃)₂ and NH₄OH (without adding pPD), and the Ni-pPD@rGO nanosheets (with the addition of pPD) has been added to **Supplementary Figs 3-5 in the revised manuscript**. These additional results serve to further validate the formation of the Ni-pPD complex. Please refer to our responses to Question 4 raised by the referee for more details.

In XPS analysis, the comparison of O 1s spectra at ~530 eV for GO and Ni-pPD@rGO has been added, as shown in the following figure (**Figure R3**), which also clearly demonstrates the reduction of GO to rGO.

Figure R3. O1s XPS spectrum of Ni-pPD@rGO nanosheets in comparison with that of GO nanosheets.

We concur with the referee's observation that there may be traces of Ni(OH)₂ in our samples. However, the predominant component is the 2D Ni-pPD complex. If Ni(OH)₂ was the primary component, the entire sample would still hydrophilic characteristics similar to GO nanosheets.^[R5] However, Fig. 2e shows that the contact angle of our samples is 84°, indicating that the main component is the 2D Ni-pPD complex on rGO nanosheets. More discussion about Ni(OH)₂ has also been provided in our responses to Question 5 raised by the referee.

Accordingly, the comparison of O 1s spectra at ~530 eV for GO and Ni-pPD@rGO have been added in Supplementary Fig. 4c in the revised manuscript.

4. Multivalent cations are known to form cross-linking with the functional groups of GO. The selectivity of the Ni²⁺ towards the GO and pPD should be studied, which is missing this work. They should compare the XPS and the IR spectra of the pure GO and GO treated with Ni(NO₃)₂ and NH₄OH, pure Ni-pPD metal complex and the Ni-pPD@rGO nanosheets.

Reply and revision: We agree with the referee's valuable comments. The first synthesis step was to disperse GO nanosheets within Ni^{2+} solutions while expelling dissolved oxygen in water. Thus, Ni^{2+} ions would be adsorbed onto the oxygenated functional groups on GO nanosheets in this step. Then, de-oxygenated pPD solutions were added together with NH_4OH to initiate the catalytic reaction between the adsorbed Ni^{2+} and pPD on top of the oxygenated functional groups. It is noted that the reaction of forming Ni-pPD can only occur at the presence of O, according to ref. 29 (previous ref. 26). **Therefore, the cross-linking effect mentioned by the referee actually played a significant role in the synthesis of Ni-pPD@rGO nanosheets. In other words, the complex of the Ni^{2+} towards GO and pPD are not two competing processes, but interactive and sequential processes.**

As suggested by the referee, we added more characterization results to compare four different types of samples, (1) pure GO, (2) GO treated with $\text{Ni}(\text{NO}_3)_2$ and NH_4OH (without adding pPD), (3) Ni-pPD@rGO nanosheets (with the addition of pPD), and (4) pure Ni-pPD metal complex. **Figure R4** of FTIR analyses show that Sample (2) of GO treated with $\text{Ni}(\text{NO}_3)_2$ and NH_4OH (without adding pPD) almost maintain the functionality of GO. No reduction of GO occurred. More importantly, N=Q=N groups, the most characteristic feature of Ni-pPD complex, are present only for Sample (3) of Ni-pPD@rGO and Sample (4) of pure Ni-pPD in FTIR spectra. **All these results suggest the participation of pPD in the reaction for the formation of Ni-pPD complex, and at the same time, for the reduction of GO to rGO.**

Figure R4. FTIR spectra of GO, $\text{Ni}(\text{OH})_2$ @GO without adding pPD, Ni-pPD@rGO nanosheets with adding pPD, pure Ni-pPD.

In **Figure R5** of XPS analysis, it is also demonstrated that, without the addition of pPD, no GO reduction occurred for Sample (2).

Figure R5. (a) C 1s and (b) O 1s XPS spectra of Ni-pPD@rGO nanosheets (with adding pPD) and Ni(OH)₂@GO (without adding pPD). (c) Comparison of Ni 2p for pure Ni-pPD, Ni-pPD@rGO nanosheets (with adding pPD) and Ni(OH)₂@GO (without adding pPD).

All in all, the conclusion is much clearer in the revised manuscript, thanks to the referee's valuable comment.

Accordingly, the following description was added in Method section on Page 21 of the revised manuscript,

“For comparison, Ni(OH)₂@GO nanosheets were also synthesized by treating GO only with Ni(NO₃)₂ and NH₄OH (without adding the pPD solution).”

In addition, the XPS and FTIR analyses of GO treated only with Ni(NO₃)₂ and NH₄OH, and pure Ni-pPD metal complex have been added in Supplementary Figs 3-5 in the revised manuscript.

5. How did the authors separate the Ni-pPD@rGO nanosheets from the bi-products formed during the reaction? The XPS analysis indicates the obtained Ni-pPD@rGO nanosheets are not in the pure form. Rather it is a mixture with Ni(OH)₂.

The reaction of Ni²⁺ ions with base forms Ni(OH)₂ nanosheets, which are insoluble in water and have positive zeta potential, which will lead to a decrease in the zeta potential.

Reply and revision: Thank you for the referee's critical comments. After the synthesis, the resulted black powders were filtered and then washed with Milli-Q water and ethanol thoroughly, in order to separate the Ni-pPD@rGO nanosheets from the possible bi-products formed during the reaction.

We concur with the referee's observation that there may be traces of Ni(OH)₂ in our samples. However, the predominant component is the 2D Ni-pPD complex. If Ni(OH)₂ was the primary component, the entire sample would still hydrophilic characteristics similar to GO

nanosheets.^[R5] However, Fig. 2e shows that the contact angle of our samples is 84°, indicating that the main component is the 2D Ni-pPD complex on rGO nanosheets. Figures R2-R7 also demonstrate that the main component is 2D Ni-pPD complex on top of rGO nanosheets.

Figure R8. Illustration of the possible defects as few oxygenated functional groups and Ni(OH)₂ in MOGMs, which would not affect water/ion permeation behavior as long as the membrane is thick enough.

Despite the presence of remaining oxygenated functional groups and Ni(OH)₂, they can be regarded as defective and ancillary structures within our membrane samples. The primary structures in the resulting Ni-pPD@rGO membranes remain as atomic-scale graphene capillaries. As long as the membrane is adequately thick, these defective structures cannot connect to form a continuous pathway. **Consequently, the atomic-scale graphene capillaries serve as the rate-limiting factor for water and ion transport across the membrane (Figure R8).** Hence, the existence of these defective structures within the membrane does not impact the discussion and conclusions presented in this study.

Accordingly, the following description was added as Supplementary Note 7 in Supplementary Information on Page 26 of the revised manuscript,

“Although there may be remaining oxygenated functional groups on rGO and traces of Ni(OH)₂ present, they can be considered as defective and secondary structures within MOGMs. The primary structures in MOGMs remain as atomic-scale graphene capillaries. As long as the membrane is sufficiently thick, these defective structures cannot form a continuous pathway. Therefore, the atomic-scale graphene capillaries continue to serve as the rate-limiting factor for water and ion transport across the membrane.”

6. As claimed in Figure 1h, the d -spacing of the Ni-pPD@rGO is 9.6 Å, which gives a free space of ~ 6.2 Å (subtracting the basal plan thickness 3.4 Å from d -spacing). Now, it is not clear how two Ni-pPD sheets each of 2.6 Å will fit between two GO flakes when the space between a GO and the Ni-pPD is 3.63 Å (Figure 1d). Even neglecting the space between two Ni-pPD monolayers, the total free space between two GO layers should be 12.46 Å.

Reply and revision: Thank you for the referee's critical comments. **Fig. 1e (previous Fig. 1d) may be a little misleading. 3.63 Å in Fig. 1e is actually the d -spacing between the rGO and the restacked Ni-pPD complex, not the free space.** This is understandable, because it is the normal π - π stacking. Therefore, three layers' π - π stacking, including rGO-pPD, pPD-pPD, and pPD-rGO, should result in the d -spacing of the restacked Ni-pPD@rGO being around $3 \times 3.63 = 10.9$ Å (**Figure R9a**). Furthermore, as also shown in Figure R9a, because of small interaction between the 1st-layer rGO and 3rd-layer pPD, rGO and pPD nanosheets will be deformed a little bit, contracting the whole d -spacing to be 9.6 Å. In this way, the free space between the 1st-layer rGO and 3rd-layer pPD is 6.01 Å - 3.4 Å = 2.61 Å, and that between the 1st-layer rGO and 4th-layer rGO is 9.58 Å - 3.4 Å = 6.18 Å (**Figure R9b**).

Figure R9. Optimized structure of restacked Ni-pPD@rGO nanosheets computed by DFT. a, Ni-pPD@rGO nanosheets before and after optimization. All values are d -spacings, not free space. **b,** The restacking results in the free space of 2.6 Å and 6.2 Å for molecules to transport.

Accordingly, we have clarified that 3.63 Å is the d -spacing between the rGO and the Ni-pPD monolayer island in Fig. 1e (previous Fig. 1d) in the revised manuscript.

7. "GOMs normally present only one X-ray diffraction (XRD) peak at around 11° (~0.8 nm) as shown in Fig. 1i." On careful observation, the peak for GOM is centered at around 10°-10.4° (0.88 to 0.85 nm).

Similarly for MOGM, the *d*-spacing is mentioned as 0.96nm, but the peak 002 is centered at around 9.5°-9.8° resulting in a *d*-spacing of around 0.93 to 0.9 nm.

Reply and revision: Thanks for the referee's careful observations. We corrected our claims to be "GOMs normally present only one X-ray diffraction (XRD) peak at around 10.4° (~0.85 nm) as shown in Fig. 2e." and "The XRD result obtained for MOGMs, however, demonstrates four distinct peaks of (002), (004), (006), and (008), with the interspacing of (002) being 0.93 nm and that of (006) being 0.32 nm, strongly supporting our conclusions about the restacked repeating unit structure of Ni-pPD@rGO nanosheets" on Page 10 in the revised manuscript.

Please note that the XRD analysis of GOMs is also dependent on the relative humidity of the environment, so that it may cause some variations. Moreover, there are four distinct peaks of (002), (004), (006), and (008) for MOGMs. We can also calculate *d*-spacings of (002) from *d*-spacings of (004), (006), and (008), which are 0.93 nm, 0.97 nm, and 0.94 nm. This XRD result clearly suggests the superlattice structure for MOGMs.

Reviewer #2:

This manuscript titled " Anomalous Liquid Gating from Atomic-scale Graphene Capillaries for Precise and Ultrafast Molecular Sieving" presents anomalous liquid gating from nanofiltration membranes based on a graphene nanoconfinement effect, fabricated from spontaneous restacking of island-on-nanosheet microstructure. The graphene nanoconfinement facilitates high-speed, diffusion-free water transport in a way analogous to Newton's cradle-like Grotthus conduction. These experiments are rationalized, and the results are sufficient to support the conclusion. Before considering this manuscript for publication, the authors should consider the following points in any revision as follows:

I will be happy to recommend for publication a revised version of the manuscript in Nature communications.

Reply and revision: We greatly appreciate the positive comments and all the very helpful suggestions from the reviewer.

1. The authors state nanoscale confined water structures have solid-like properties that hinder diffusion under osmosis. It is not the liquid we usually know. Basically, liquid gating technology is to use the liquid as a mechanical gate to control the transport of fluid. Therefore, the liquid gating phenomenon described in this manuscript is not appropriate. The title needs to be reconsidered. For example, using "water molecular gating".

Reply and revision: We greatly appreciate the referee's important comment and helpful suggestion. The phrase "water molecular gating" indeed grasps the essence of this work. Thus, following the suggestion, we have changed the phrase of "liquid gating" to "water molecular gating" in the revised manuscript.

2. The introduction is not well arranged, and the logic is not clear. And the use of many conjunctions is confusing. For example, there is no causal relationship between " As such, ineffective ion sieving properties have been found on reduced graphene oxide membranes" and the previous sentence.

Reply and revision: We really appreciate the referee's valuable comments. Following the helpful suggestion, we have reorganized the introduction in the revised manuscript. We hope this version would be much easier for readers to understand.

3. AFM images of the nanosheet thickness before restacking are encouraged to be provided for better comparison.

Reply and revision: We really appreciate the referee for raising this critical and valuable comment. Unfortunately, we cannot obtain the nanosheet before restacking. It is the spontaneous restacking after the catalytic reaction, similar to the spontaneous restacking of rGO nanosheets after GO reduction. However, the referee indeed reminds us to check the fine structure of Ni-pPD monolayers on top of the restacked Ni-pPD@rGO nanosheets. Please see the following

figure (**Figure R10**), which is consistent with HAADF image in Fig. 1c in the revised manuscript.

Figure R10. AFM analysis showing monolayer island structure on top of the spontaneously restacked Ni-pPD@rGO nanosheets. **a**, AFM image. **b**, Height profile. The fine monolayer island structure has been highlighted by pink circles.

Accordingly, the above figure has been added as Supplementary Figure 8 in the revised manuscript.

4. Although the optimized structure is shown in the paper (Fig. 1d), it is necessary to give the initial structure before optimization in SI.

Reply and revision: We really appreciate the referee's comments. The initial structure has been provided in the revised manuscript, as also shown below.

Figure R11. The initial structure of Ni-pPD@rGO nanosheets before relaxation. **a,b**, view from two different angles.

Accordingly, the above figure has been added as Supplementary Figure 7 in the revised manuscript. In addition, the following description has been added on Page 6 of the revised manuscript,

“As compared with the heterostructure before relaxation (Supplementary Fig. 7), Fig. 1e demonstrates that the four benzene rings around Ni ions in Ni-pPD tend to be planarized by rGO through π - π stacking.”

5. For Fig. 1h, please explain further why does the restacked structure provide distinct free space is provided? And mark the area corresponding to free space in the figure.

Reply and revision: We really appreciate the referee to point this out. As shown in **Figure R12a,b** (Supplementary Fig. 9 in the revised manuscript), the d -spacing between the 1st-layer rGO and 3rd-layer pPD is 6.0 Å, giving the free space of 2.6 Å; and also, the d -spacing between the 1st-layer rGO and 4th-layer rGO is 9.6 Å, giving the free space of 6.2 Å. The area corresponding to free space has been marked in Fig. 2d in the revised manuscript, which is also shown in **Figure R12c**.

Optimized Restacked Structure

Figure R12. Optimized structure of restacked Ni-pPD@rGO nanosheets computed by DFT. a, All labels are interlayer spacings. **b,c** The restacking results in the free space of 2.6 Å and 6.2 Å for molecules to transport.

Accordingly, Fig. 2d (previous Fig. 1h) has been redrawn in the revised manuscript. In addition, the following descriptions have been added on Page 8 in the revised manuscript,

“As such, the d -spacing between the 1st-layer rGO and 3rd-layer pPD was calculated to be 6.0 Å, giving the free space of 2.6 Å (marked as pink slash lines in Fig. 2d); and also, the d -spacing between the 1st-layer rGO and 4th-layer rGO was determined to be 9.6 Å, giving the free space of 6.2 Å (marked as blue slash lines in Fig. 2d).”

6. In the sentence “We examined an unexpected permeation behavior on MOGMs (Fig. 2a)”, the Fig. 2a did not show the unexpected behavior. And abbreviations should be defined when they first appear in the text. Write the full name and give the abbreviation in parentheses.

Reply and revision: We really appreciate the referee to point this out. The sentence has been corrected on Page 11 of the revised manuscript to be,

“We examined an unexpected permeation behavior on MOGMs (Fig. 2b).”

In addition, we corrected the definition of the abbreviation on Page 9 of the revised manuscript to be

“These new membranes are, therefore, denoted as metal-organic (Ni-pPD) pillared, graphene-based membranes (MOGMs).”

7. The caption of Fig. 2c was unclear, does the upper subgraph in Fig. 2c represent microscopic states at different times? And the explanation for Figure 2c is presented in an untimely manner, appearing only in the concluding discussion section. This layout lacks coherence and may not be considered optimal.

Reply and revision: We agree with the referee’s valuable comments. According to the referee’s suggestion, we have moved Fig. 2c and the corresponding discussion to the end of the revised manuscript. In addition, Fig. 2c has been modified to be a new figure of Fig. 6 in the revised manuscript, which was shown as **Figure R13** in our responses to Question 12.

8. In the sentence “The membranes made of the restacked Ni-pPD@rGO nanosheets have then been fabricated by vacuum filtration, followed by the controlled evaporation under external pressures.”, What does it mean to control evaporation under external pressure? Does it have any special effect to the fabricating process?

Reply and revision: We really appreciate the referee to point this out. As described in Methods and also illustrated in Supplementary Figure 10, after vacuum filtration, the membranes were immediately clamped between two glass plates and dried under vacuum at 70°C. The control of evaporation is an important step for the synthesis of MOGMs with well-defined, atomic-scale graphene capillaries at 6 Å. Too fast or uneven evaporation will cause the formation of large pores, in which case the anomalous water molecular gating mechanism would not be observed.

To better clarify this point, the following description has been added to the Method section on **Page 22 of the revised manuscript**,

“The controlled evaporation is an essential step to make sure MOGMs with well-defined, atomic-scale graphene capillaries at 6 Å.”

9. How durable and resistant to staining is the membrane?

Reply and revision: We really appreciate the referee's valuable comments. The referee raised a very critical comment. This study is mainly to discover the anomalous water molecular gating behavior and its underlying mechanism. In addition, the durability and resistance to staining can also be improved by the design of a composite membrane based on the current MOGMs. Thus, we intend to explore this avenue in our future research and hope to report on it in the near future.

10. The contact angle of the MOGMs is 84° . How does its hydrophilic or hydrophobic nature affect the anomalous gating behavior?

Reply and revision: We really appreciate the referee's valuable comments. The hydrophilic or hydrophobic nature of the atomic-scale capillaries is essential in activating the anomalous water molecular gating behavior. The presence of oxygenated functional groups would disturb the formation of 2D crystal-like, hydrogen-bonded water network within the atomic-scale capillaries. Thus, this anomalous permeation phenomenon has not been observed on hydrophilic GOMs in previous reports. On the other hand, we believe that the atomic-scale hydrophobic capillaries (contact angle $> 90^\circ$) would not generate this anomalous permeation phenomenon either, since it is difficult for water molecules to enter the hydrophobic capillaries. Therefore, atomic-scale, graphene capillaries is the key for the anomalous gating behavior.

According to our calculations, the fundamental reason is the attractive van der Waals interactions between the graphene walls and water molecules. Although it is relatively weak in large capillaries, this short-range attraction plays a key role within the atomic-scale graphene capillaries and a large capillary pressure can be generated (in the order of 1,000 bar). Therefore, the contact angle of 84° for MOGMs, which is close to the contact angle of water on graphite (85°),^[R3] is one of the most important characteristics in activating the anomalous water molecular gating behavior reported in this work.

11. The water flux keeps increasing up to $2.5 \text{ L m}^{-2} \text{ h}^{-1}$ at $9.8 \times 10^{-3} \text{ bar}$ ("ON" state in Fig. 2b). If the hydrostatic pressure is continuously increased, will the water flux rate continue to increase? Is there a turning point hydrostatic pressure?

Reply and revision: We really appreciate the referee's valuable comments. As displayed in Fig. 5d in the revised manuscript, the water flux rate will continue to increase if the hydrostatic pressure increases further. In addition, the referee raised a very interesting topic about the turning point for the continuous increase of the hydrostatic pressure. It requires the systematic investigation about the extremes of the newly-found anomalous permeation phenomenon. This work is undertaken and we hope the related work will be reported in the near future.

12. This work is interesting, but it lacks an appropriate schematic diagram to clearly explain its mechanism, making it difficult for readers to understand easily. It is better to modify Fig. 2 to help reader better understand the mechanism of the anomalous gating behavior.

Reply and revision: We really appreciate the referee's valuable comments. We have modified Fig. 2c and moved it to a new figure of Fig. 6 at the end of the revised manuscript. In Fig. 6 (also

shown in **Figure R13**), schematic diagrams have been demonstrated to clearly explain its mechanism; especially, Newton’s cradle-like Grotthus Conduction for fast water flux.

Figure R13. Microscopic illustrations of water molecular gating mechanism. a, “OFF” state. **b**, “ON” state with Newton’s cradle-like Grotthus Conduction originated from highly-connected hydrogen-bonded network.

The new Fig. 6b (previous Fig. 2c) represent two continuous states for the collision of one extra molecule onto the 2D ordered water nanosheets, leading to almost instant transport of the water molecule to the other side of the capillary. This is analogous to Newton’s cradle-like Grotthus Conduction. **More importantly, the two continuous states in Fig. 6b together represent only one snapshot for each collision.** In real operations, it should be continuous collision of water molecules from the left, resulting in instant but continuous water transport to the other side of the capillary. **Thus, the continuous mode of this analogy forms bulk flow of water.**

13. For the DFT simulation, what are the criteria for hydrogen bonding between water molecules? In addition, the role of Ni-pPD is not reflected in the model. Please explain in detail the special role of Ni-pPD in this system and the significant difference in properties between rGO nanosheets without Ni-pPD.

Reply and revision: We really appreciate the referee’s valuable comments. As inspired by the referee, we put the quantitative analysis of the average number of hydrogen bonds in the revised manuscript in Supplementary Fig. 20.

As well accepted in the literature, the definition for a hydrogen-bond is that the distance between two oxygen atoms $R_{O-O} < 3.5 \text{ \AA}$ and $\alpha < 30^\circ$.^[R6] As shown in Figure R14, putting these criteria into VMD software, the average number of hydrogen bonds for the structure in **a** can be

calculated to be 44 in **b**. Considering the hexagonal type of unit cell, there are around 18.2 water molecules in **a**. In addition, each hydrogen bond is shared by two water molecules, and thus, there are averagely 4.83 hydrogen bonds around each water molecule ($\text{HB} \cdot \text{H}_2\text{O} = 4.83$).

Figure R14. Quantitative determination of the average number of hydrogen bonds around each water molecule. **a**, The water structure within the atomic-scale graphene capillaries. **b**, The calculation of the average number of hydrogen bonds in **a** using VMD software.

Ni-pPD monolayer islands were used as pillars between two rGO nanosheets to create atomic-scale graphene capillaries. If there is no Ni-pPD, two rGO nanosheets would restack and seal the atomic-scale graphene capillaries. In addition, the Ni-pPD monolayer islands is mainly composed of the graphitic structure. Thus, the role of Ni-pPD is not reflected in the model.

Accordingly, the above figure has been added as Supplementary Figure 20 in the revised manuscript.

14. In the sentence “the energy barrier was obtained by subtracting the energy before the move from that after the move” of supplementary Note3, does the energy mean all energy in the system?

Reply and revision: The energy is the system energy, but the description was not correct in the original manuscript. We really appreciate the referee to point this out.

Accordingly, we changed the description in Supplementary Note 3 in the revised manuscript as follows:

“Each point in **b** is the system energy calculated, while the energy barrier can be obtained from the maximum energy value along the designated transport route.”

15. For Fig. S20, is there any quantitative correlation between the On/Off Switch Height and the hydrating ion radius of different ions?

Reply and revision: We really appreciate the referee's valuable comments. The referee raised a very interesting point that the On/Off switch height can be correlated to the hydrating ion radius of different ions. We have the same perspective as the referee suggested. However, we noticed that the error bar of On/Off switch height is relatively large, so that the quantitative correlation may not be as accurate as we expected. Therefore, this point has been qualitatively explained in our supplementary note 5 in the supporting information.

16. Please change the term "Van der Waals" to "van der Waals".

Reply and revision: We really appreciate the referee to point this out. We have changed the term "Van der Waals" to "van der Waals" in our revised manuscript.

17. According to the theoretical computations, the "ON" state shows the bulk movement of water molecules, which seems not the same as the movement of Newton's cradle-like Grotthus conduction. Authors should consider whether this description is appropriate or not.

Reply and revision: We really appreciate the referee's valuable comments. **Actually, we can see the bulk movement of water molecules in the "ON" state to be the continuous mode of Newton's cradle-like Grotthus Conduction.** The new Fig. 6b (previous Fig. 2c) represent two continuous states for the collision of one extra molecule onto the 2D ordered water nanosheets, leading to almost instant transport of the water molecule to the other side of the capillary. This is analogous to Newton's cradle-like Grotthus Conduction. More importantly, the two continuous states in Fig. 6b together represent only one snapshot for each collision. In real operations, it should be continuous collision of water molecules from the left, resulting in instant but continuous water transport to the other side of the capillary. Thus, the continuous mode of this analogy forms bulk flow of water.

Although the highly hydrogen-bonded network of water molecules formed within the atomic-scale graphene capillaries would not be as solid as those cradle balance balls, but we believe this analogy grasp the essence of the almost instant transport of water molecules from one side to the other side through the capillary.

Accordingly, we keep this analogy and add more descriptions on Page 19 of the revised manuscript to be,

"The bulk movement of water molecules in the "ON" state is then the outcome of the continuous mode of Newton's cradle-like Grotthus Conduction."

18. The shaded and solid sections of the bar chart in Fig 4c should indicate the meaning clearly.

Reply and revision: We really appreciate the referee's valuable comments. Fig. 5c (previous Fig. 4c) has been redrawn as shown in **Figure R15** for clear presentation.

Figure R15. Ultrafast water flux and effective ion sieving achieved simultaneously for various ions on MOGMs.

19. *How does the graphene capillaries size influence ion diffusion and water transport.*

Reply and revision: We greatly appreciate the referee's valuable comments. This is also a very interesting point raised by the referee. While it remains challenging to experimentally adjust the size of graphene capillaries beyond 0.6 nm at the atomic scale, we can explore the influence of graphene capillary size on ion diffusion and water transport *via* simulations. Fig. 4c shows the graphene capillaries with the *d*-spacing of 0.6 nm and Supplementary Figure 19 shows the graphene capillaries with the *d*-spacing of 0.87 nm. The former would form one layer of 2D crystal-like water nanosheet, while the latter would form two layers of 2D crystal-like water nanosheets.

In Supplementary Figure 22 and Supplementary Movies 3 and 4, we have further demonstrated the influence of the size of graphene capillaries on ion diffusion and water transport (see the Supplementary Note 4). Increasing the graphene capillaries further over the *d*-spacing of 1 nm, the structure of the included water molecules starts to transit from 2D crystal-like water nanosheet to bulk liquid water. In this case, the graphene nanoconfinement effect would vanish accordingly.

20. *The authors are suggested to check through the manuscript to improve the English. It is difficult to read.*

Reply and revision: We really appreciate the referee to point this out. We have reorganized the whole manuscript and improve the English in the revised version. Hope this version would be clear enough for readers to grasp the highlights and essence of our work easily.

21. Many recent publications related to ion transport behavior, such as Nat Commun 2022, 13, 6709, Nano Letters 2020, 20: 6937–6946, Joule 2023, 7: 251–253. In order to help readers better understand the importance of this work, these references should be cited.

Reply and revision: We really appreciate the referee's valuable comments. These references are indeed very important and we added them in our revised manuscript as references 15, 24, and 25.

Reviewer #3:

This manuscript describes a clever catalytic synthesis method to fabricate a sub-nano-confined, graphene-based microstructure. Ions can be selectively sieved by gating water molecules on/off. The mechanism is presented that water forms crystal-like water nanosheets inside the graphene capillaries and the application of a small hydrostatic pressure causes the water nanosheet to slide as a whole, enabling fast transport. This is very interesting. In addition, combing the experimental data with MD simulation has greatly improved the quality of this work. The manuscript can be considered for publications after minor revisions, as the comments listed below.

Reply and revision: We greatly appreciate the positive comments and all the very helpful suggestions from the reviewer.

1. How to make sure there is only one layer of Ni-pPD on top of rGO in assisting the formation of sub-nano-confined graphene capillaries?

Reply and revision: Thank you for the referee's valuable comment. The most significant evidence for this is BET and pore size distribution results shown in Fig. 2f and Supplementary Fig. 13 in the revised manuscript. The strongest peak in Fig. 2f sits at the pore size of 6.0 Å, although the pores smaller than 4.5 Å cannot be detected by CO₂ gas. The pores with the size of 6.0 Å can only be formed in the case that there is only one layer of Ni-pPD on top of rGO, which corresponds well with XRD, TEM and the simulation results in Figs 1 and 2.

Two other peaks below 1 nm in Fig. 2f imply that more than one layer of Ni-pPD may exist on top of rGO. However, because those peaks are relatively weak compared with the main peak at 6.0 Å, it is confident that most nano-islands on rGO nanosheets are monolayer Ni-pPD. Also, because the smallest pore size is the determining factor for molecular transport through the macroscopic-scale membranes, the discussion and conclusions made in the main text would not be affected by the pores with large sizes.

Figure R10. AFM analysis showing monolayer island structure on top of the spontaneously restacked Ni-pPD@rGO nanosheets. a, AFM image. b, Height profile. The fine monolayer island structure has been highlighted by pink circles.

To further confirm the monolayer of Ni-pPD on top of rGO, we also checked the fine structure of Ni-pPD monolayers on top of the restacked Ni-pPD@rGO nanosheets by AFM analysis. Please see the above figure (**Figure R10**), which is consistent with HAADF image in Fig. 1c in the revised manuscript.

Accordingly, the above figure has been added as Supplementary Figure 8 in the revised manuscript. The following descriptions have been revised in Supplementary Note 1 on Page 14 in the Supporting Information to be,

“The strongest peak sits at the pore size of 6.0 Å, corresponding well with XRD and the simulation results, although the pores smaller than 4.5 Å cannot be detected by CO₂ gas. Two other peaks below 1 nm in Fig. 2f may imply that some sub-nano graphene capillaries are formed with three or four layers of Ni-pPD nano-islands sandwiched between. However, because those peaks are relatively weak compared with the main peak at 6.0 Å, it is confident that most nano-islands on rGO nanosheets are monolayer Ni-pPD. Also, because the smallest pore size is the determining factor for molecular transport through the macroscopic-scale membranes, the discussion and conclusions made in the main text would not be affected by the pores with large sizes.”

2. *The process of ON/OFF is unclear. The authors should clearly define how the pressure is applied.*

Reply and revision: Thank you for the referee’s valuable comments. The hydrostatic pressure was applied by increasing the level (height) of the water on the feed (water side) compartment. Over the threshold pressure, the ultrafast water flux will be initiated.

Accordingly, the following description has been added on Page 12 of the revised manuscript,

“More importantly, water flow can be turned on and off reversibly (Fig. 3b), by adjusting hydrostatic pressures higher or lower than the gating pressures (P_G : $\sim 10^{-2}$ bar), revealing the successful establishment of a new gating mechanism for precise and ultrafast molecular sieving.”

3. *The membrane is not a single capillary, how to rationalize the simulation with only one graphene capillary without considering the tortuous pathways along ~2.3-micron thick graphene membranes?*

Reply and revision: We really appreciate the referee’s valuable comments. It is true that the ~2.3-micron thick graphene membranes have large tortuosity where the water flow will also be influenced by the edge carbon atoms, dangling bonds and hydrogen bonding between the functional groups; **however, the key structure to generate frictionless water flow in MOGM membranes is the atomic-scale graphene capillaries.** As discussed in our manuscript, the edge carbon atoms, dangling bonds and hydrogen bonding between the functional groups **would not induce the frictionless flow of water molecules but provide resistance to the water flow.**

This resistance can be attributed to the gating pressures (or energy barriers) that we need to overcome in initiating ultrafast water flux (Supplementary Fig. 25).

Accordingly, the following descriptions have been revised on Pages 19 and 20 in revised manuscript to be,

“Therefore, although the micrometer-thick MOGMs have long and torturous permeation paths, it is this high-speed, Grotthus conduction that enables ultrafast water flux.”

“In addition to the edge carbon atoms, dangling bonds, and the interstitial water/ions outside of the graphene capillaries that could add resistance to the bulk water flow, Supplementary Fig. 23 also implies that the counter motion of ions, although suppressed, also exert resistance to the sliding of the ordered water nanosheets. All these factors can contribute to the magnitude of gating pressures (Supplementary Fig. 25).”

4. *The resolution in several figures is not high enough. For example, Fig. 1b.*

Reply and revision: Thank you for the referee’s valuable comments. We tried many advanced characterization techniques for several times, but still, it is very challenging to obtain the atomic resolution of the heterostructure with the following reasons. First, in STEM or HAADF, our samples are moving under the electron beam with high accelerating voltages. Second, the monolayer of Ni-pPD itself is in amorphous structure. Third, C atoms are light elements and there is only one monolayer of Ni-pPD on top of rGO. The signals from these two layers would interact with each other, preventing us from clear demonstration of the atomic resolution of the heterostructure.

Although the atomic resolution of the heterostructure is difficult to obtain, the main features of the heterostructure can be determined clearly. In addition to HRTEM and HAADF characterization, we also detected distinct features from XRD and BET/Pore size distribution, all suggesting that the heterostructure is composed of Ni-pPD monolayer-islands distributed uniformly on rGO nanosheets. DFT calculations were further conducted to understand the microstructures and how the heterostructure formed, which correspond well with the experimental results.

5. *How is the comparison between the DFT simulation results and experimental results?*

Reply and revision: We really appreciate the referee’s valuable comments. DFT simulation results present in our work agree well with our experimental results.

First, the optimized structure computed by DFT (Supplementary Fig. 9) demonstrates that two layers of Ni-pPD are pillared between two rGO through π - π stacking with all the stacking distances being in the range from 0.34 to 0.36 nm, generating the interlayer spacing of 0.96 nm from the top to the bottom. This result agrees well with the XRD result in Fig. 2e and HRTEM result in Fig. 2c in the revised manuscript.

Second, as shown in the simulated results in Fig. 2d, the d -spacing between the 1st-layer rGO and 3rd-layer pPD was calculated to be 6.0 Å, giving the free space of 2.6 Å (marked as pink slash lines in Fig. 2d); and also, the d -spacing between the 1st-layer rGO and 4th-layer rGO was determined to be 9.6 Å, giving the free space of 6.2 Å (marked as blue slash lines in Fig. 2d). This result corresponds well with the BET/Pore size distribution result, since Fig. 2f clearly demonstrates three sharp and distinctive peaks below 1 nm, with the strongest peak sitting at ~0.6 nm.

Third, the optimized arrangements of water molecules within 6.0 Å-interspaced graphene capillary in Fig. 4c show a monolayer of 2D ordered structure. The average number of hydrogen bonds around each water molecule has been calculated to be $\text{HB}\cdot\text{H}_2\text{O} = 4.83$ in Supplementary Fig. 20. The in-situ FTIR shows the increased intensity of blue and cyan peaks for MOGMs in 37% humidity, which correspond to strongly-bonded water network included with $\text{HB}\cdot\text{H}_2\text{O} > 4$. Thus, DFT simulation results are also consistent with the experimental results.

Based on these, all molecular dynamics (MD) simulations conducted in this work reflect the real case of our observations in the permeation test, which gives us deep understanding of the anomalous water molecular gating mechanism. Therefore, all our DFT simulation results agree well with our experimental results.

6. The flux data on Figure S13 extrapolate to zero LMH at zero applied hydrostatic pressure, while for the data on the Figure S12, the data extrapolate to about 4LMH. What is the reason for such a difference?

Reply and revision: Thank you for the referee's valuable comments. We think the referee raised a very interesting point that we did not pay attention to. When extrapolating the water flux data to zero applied hydrostatic pressure, Supplementary Fig. 16 (previous Supplementary Fig. 13) shows zero LMH for 0.1 M KCl draw solution, while Supplementary Fig. 15 (previous Supplementary Fig. 12) shows 4 LMH for 0.1 M NaCl draw solution. This observation means that the slope of the water flux data in the "ON" state is much smaller for the case of 0.1 M NaCl than that for the case of 0.1 M KCl. **This is actually another experimental proof that the resistance for the sliding of the 2D water nanosheet to the draw (salt side) is dependent on the type of the draw solutions.** In other words, the resistance is higher for the draw solution of 0.1 M NaCl than that for the draw solution of 0.1 M KCl. We really appreciate the referee's thoughtful considerations.

Accordingly, the following descriptions have been added into Supplementary Note 5 on Page 24 in Supporting Information to be,

"When extrapolating the water flux data to zero applied hydrostatic pressure, Supplementary Fig. 16 shows zero LMH for 0.1 M KCl draw solution, while Supplementary Fig. 15 shows 4 LMH for 0.1 M NaCl draw solution. This observation means that the slope of the water flux data in the "ON" state is smaller for the case of 0.1 M NaCl than that for the case of 0.1 M KCl. This is also

another experimental proof that the resistance for the sliding of the 2D water nanosheet to the draw (salt side) is dependent on the type of the draw solutions.”

7. *The authors refer to the nanosheets as graphene at times which is not the correct use of terminology. The authors should choose and use the appropriate term.*

Reply and revision: We greatly appreciate the valuable comments. The terminology has been corrected to be “nanosheets or rGO” in the revised manuscript. The term “graphene” has only been used to describe the capillaries, which are atomic-scale graphene capillaries.

8. *There are several errors in references formats.*

Reply and revision: We really appreciate the referee’s valuable comments. We have corrected the errors of the references’ formats in the revised manuscript. We would like to thank the referee again for the careful review of these typos.

References:

- [R1] Liu, X. et al. 2D material nanofiltration membranes: from fundamental understandings to rational design. *Adv. Sci.* **8**, 2102493 (2021).
- [R2] Wilson, N. R. et al. Graphene oxide: structural analysis and application as a highly transparent support for electron microscopy. *ACS Nano* **3**, 2547–2556 (2009).
- [R3] Radha, B. et al. Molecular transport through capillaries made with atomic-scale precision. *Nature* **538**, 222–225 (2016).
- [R4] Liu, H. Y., Wang, H. T. & Zhang, X. W. Facile fabrication of freestanding ultrathin reduced graphene oxide membranes for water purification. *Adv. Mater.* **27**, 249–254 (2015).
- [R5] Zheng, S. X., Tu, Q. S., Urban, J. J., Li, S. F. & Mi, B. X. Swelling of graphene oxide membranes in aqueous solution: characterization of interlayer spacing and insight into water transport mechanisms. *ACS Nano* **11**, 6440–6450 (2017).
- [R6] Liu, J., He, X., Zhang, J. Z. H. & Qi, L. W. Hydrogen-bond structure dynamics in bulk water: insights from *ab initio* simulations with coupled cluster theory. *Chem. Sci.* **9**, 2065–2073 (2018).

Reviewer(s)' Comments to Author:

Reviewer #1:

Authors have revised the manuscript significantly, especially to address the synthesis and characterization of the membranes. They have addressed most of the queries, however the following remain to be answered. I would suggest them to address the below before their manuscript can be accepted.

Reply and revision: We greatly appreciate the valuable comments and all the very helpful suggestions from the reviewer.

1. In the abstract, it is mentioned, “.....graphene capillaries at 6 Å, which were.....” However, in Figure 2d and Supplementary Figure 9, it is mentioned as 6.2 Å. In view of this, the authors can put an error and describe it. Perhaps, does the Ni-pPD@rGO membrane expand in water?

Reply and revision: Thank you for the referee’s valuable comments. Graphene capillaries with the size of ~6 Å were determined experimentally through pore size distribution (PSD) tests of MOGMs as shown in Figure 2f. However, the size of 6.2 Å was calculated from DFT analyses. In experiment, it is very hard to detect 0.2 Å difference in pore size, and thus, we claim that the result from DFT calculations corresponds well with our experimental investigations. Furthermore, this little difference has no relation to the membrane expansion in water.

However, we agree with the referee that this 0.2 Å difference may cause confusions to the readers. Accordingly, we added the following comment on Page 10 of the Revised Manuscript as follows:

“Fig. 2f clearly demonstrates three sharp and distinctive peaks below 1 nm, with the strongest peak sitting at ~0.6 nm. Considering the calculated free space of 6.2 Å in Fig. 2d, the PSD experimental results in Fig. 2f correspond well with our DFT calculations.”

2. Regarding the zeta potential of the GO and the Ni-pPD@rGO, please describe in brief how they are measured. For instance, what was the concentration of the nanosheets dispersion. Also, please include the zeta potential of the Ni-pPD complex in Supplementary Fig. 6.

Reply and revision: Thanks for the referee’s critical comments. To investigate the change of surface charge before and after the designated reaction, the GO and the Ni-pPD@rGO were dispersed in Milli-Q water, respectively, forming the colloidal solutions at the concentration of 0.05 mg/ml for zeta potential tests.

Accordingly, the detailed experimental procedures of zeta potential tests have been added in the experimental section on Page 23 of the Revised Manuscript as follows:

“Zeta potential was performed on Malvern zeta sizer nano series, with Ni-pPD@rGO nanosheets, GO nanosheets, and pure Ni-pPD particles being dispersed in Milli-Q water in the concentration of 0.05 mg/ml.”

As also suggested by the referee, the zeta potential of the Ni-pPD complex has been detected using the same procedure. This result has been added into Supplementary Fig. 6 on **Page 7 of Supporting Information in the Revised Manuscript**, which is also shown as Figure R1 below:

Figure R1. Zeta potential analyses of Ni-pPD@rGO nanosheets in comparison with GO nanosheets and pure Ni-pPD particles.

3. In response to the previous question in review about the free space, “As claimed in Figure 1h, the d -spacing of the Ni-pPD@rGO is 9.6 \AA , which gives a free space of $\sim 6.2 \text{ \AA}$ (subtracting the basal plan thickness 3.4 \AA from d -spacing). Now, it is not clear how two Ni-pPD sheets each of 2.6 \AA will fit between two GO flakes when the space between a GO and the Ni-pPD is 3.63 \AA (Figure 1d). Even neglecting the space between two Ni-pPD monolayers, the total free space between two GO layers should be 12.46 \AA .”

The authors have replied “Fig. 1e (previous Fig. 1d) may be a little misleading. 3.63 \AA in Fig. 1e is actually the d -spacing between the rGO and the restacked Ni-pPD complex, not the free space. This is understandable, because it is the normal π - π stacking. Therefore, three layers’ π - π stacking, including rGO-pPD, pPD-pPD, and pPD-rGO, should result in the d -spacing of the restacked Ni-pPD@rGO being around $3 \times 3.63 = 10.9 \text{ \AA}$ (Figure R9a). Furthermore, as also shown in Figure R9a, because of small interaction between the 1st-layer rGO and 3rd-layer pPD, rGO and pPD nanosheets will be deformed a little bit, contracting the whole d -spacing to be 9.6 \AA . In this way, the free space between the 1st-layer rGO and 3rd-layer pPD is $6.01 \text{ \AA} - 3.4 \text{ \AA} = 2.61 \text{ \AA}$, and that between the 1st-layer rGO and 4th-layer rGO is $9.58 \text{ \AA} - 3.4 \text{ \AA} = 6.18 \text{ \AA}$ (Figure R9b).”

However, it is not clear but is more confusing. If there were two Ni-pPD islands in between the rGO, there would be almost no space (about 1 angstrom) as seen in supplementary Figure 9. In fact with this calculation, one can say that graphene capillaries are conducting only along the regions devoid of the Ni-pDP as illustrated in the revised figure 2g. The top view schematic is

clearer however the cross-sectional schematics do not convey the correct spacings, especially as the numbers do not add up (Supplementary Figure 9, and figure 2d). In principle rGO should not have free space enough for water transport, however with their Ni-pDP pillars the authors have sufficiently maintained the rGO distance to allow water transport. This message is not conveyed clearly and is lost in the details.

In the revised manuscript on page 8, line 148 onwards, the authors wrote “Considering a monolayer of Ni-pPD nano-islands formed on each side of rGO, the repeating unit of the restacked structure between two rGO should be around 1 nm”. How is this 1 nm gotten?

If there are two monolayers of Ni-pPD between two rGO sheets), **the channels should be blocked at the interface of the two monolayers of Ni-pPD** (as shown in Fig 2d and Supplementary Figure 9). This is not clearly conveyed.

There is a lack of proper explanation, which needs to be rectified. A proper schematic is required to show the nanofluidic channels between two rGO sheets (like in Figure 2g). The present schematic is confusing as it is showing the channels are blocked. What is the pathway of the ion transport, along the plane or perpendicular to the plane of the rGO?

Reply and revision: Thank you for the valuable comments. We agree with the referee that “one can say that graphene capillaries are conducting only along the regions devoid of the Ni-pDP”. Those graphene capillaries are the ones with the size of $\sim 6.0 \text{ \AA}$ (Figure R2a).

The referee mentioned that “If there were two Ni-pPD islands in between the rGO, there would be almost no space”. This would be the case when two Ni-pPD-island nanolayers match their size and position exactly between two rGO nanosheets (Figure R2a). However, when the two Ni-pPD-island nanolayers do not match their size and position (Figure R2b), there would be another type of graphene capillaries created at around 2.6 \AA , according to our DFT calculations about the cross-sectional view of the restacked structure in Figure 2d.

Figure R2. Cross-sectional illustration of two kinds of the restacking situations with two monolayers of Ni-pPD nano-islands between two rGO nanosheets.

Therefore, in the top view schematic Figure 2g, there are three kinds of regions between two rGO nanosheets. One type contains two monolayers of Ni-pPD, which, as also mentioned by the referee, blocks the channel completely. The second type is the region free of any Ni-pPD islands, resulting in the graphene capillaries at $\sim 6 \text{ \AA}$. There is a third type of region, where only one monolayer of Ni-pPD exists within two rGO nanosheets, resulting in graphene capillaries at 2.6 \AA for molecular transport.

According to the referee's suggestion, the top view schematic in Figure 2g (as also shown in Figure R3) was redrawn to show the fully-pillared (blocked) region by pink-shaded frames, leaving 2.6 Å and 6.0 Å graphene capillaries for molecules to transport. Therefore, the pathway of the water/ion transport is along the plane of the rGO, labelled by the red arrows.

Figure R3. Schematic of the sub-nano pathways to conduct water molecules in MOGMs. Green: bottom Ni-pPD nano-islands; Yellow: top Ni-pPD nano-islands; Pink-shaded frames: fully blocked regions by the restacking of two Ni-pPD nano-islands.

The case that the numbers do not add up in Supplementary Figure 8 (previous Supplementary Figure 9) and Figure 2d is due to the distortion of Ni-pPD monolayers within two rGO nanosheets, which is clearly presented in Supplementary Figure 8a (also in Figure R4). This phenomenon is also caused by the mismatch of the two Ni-pPD monolayers, in which case the 4th layer rGO will attract the 2nd layer Ni-pPD to some extent.

Figure R4. Cross-sectional illustration (a) and DFT calculated result (b), showing the distortion of the Ni-pPD monolayer caused by the attraction between the 2nd layer Ni-pPD and the 4th layer rGO.

We also agree with the referee that “In principle rGO should not have free space enough for water transport, however with their Ni-pDP pillars the authors have sufficiently maintained the rGO distance to allow water transport. This message is not conveyed clearly and is lost in the details.” Accordingly, we added this sentence on Page 8 in the revised manuscript to emphasize the key message we tried to deliver in this work.

“Therefore, although the restacked rGOs should not present free space, the rGOs pillared by Ni-pDP nano-islands can sufficiently maintain their distance for molecular transport.”

Since Ni-pPD is mainly composed of four benzene rings in a plane on top of rGO, the theoretical molecular size of the Ni-pPD nano-islands is similar to that of a monolayer of graphene, 3.4~3.6 Å. In this case, there are three layers of π - π stacking for the repeating unit of the restacked structure between two rGO. They are, 1) the 1st layer rGO with benzene rings in the 2nd layer Ni-pPD, 2) benzene rings in the 2nd layer Ni-pPD with benzene rings in the 3rd layer Ni-pPD, 3) benzene rings in the 3rd layer Ni-pPD with the 4th layer rGO, resulting in $0.34\sim 0.363\text{ nm} \times 3 = 1.02\sim 1.09\text{ nm}$.

Why the 002 peak of the MOGM in Figure 2e is very wide? Does it indicate the presence of rGO-pPD-rGO along with rGO-pPD-pPD-rGO orientation?

We totally agree with the referee’s understanding. It is the case of “rGO-(Ni-pPD)-rGO” that causes the distortion of Ni-pPD monolayers within two rGO nanosheets, leading to the numbers that do not add up in Supplementary Figure 8 and the graphene capillaries at 2.6 Å in Figure 2d.

4. “The pore size distribution (PSD) analysis was conducted using the slit pore geometry” How the sample were prepared for this experiment? Is it in MOGMs form? Please comment on the pore size of 0.87 nm and 1 nm, shown in Fig 2f.

Reply and revision: The referee is right. The samples prepared for pore size distribution (PSD) tests are in their freestanding MOGMs form, in order to detect the graphene capillaries formed after the restacking of Ni-pPD@rGO nanosheets.

Accordingly, the detailed experimental procedures of PSD tests have been added in the experimental section **on Page 23 of the Revised Manuscript** as follows:

“Gas-adsorption measurements were made using Micromeritics ASAP 2020 using N₂ gas at 77K as well as carbon dioxide at 273.2 K. Before the gas-adsorption experiment, freestanding MOGMs were thoroughly washed and dried, and then “outgassed” under vacuum at 150 °C for 8 h.”

In addition, according to the referee’s valuable suggestion, the comments regarding the pore size of 0.87 nm and 1 nm shown in Figure 2f have been added **on Page 13 of Supporting Information in the Revised Manuscript**, as follows:

“**Supplementary Note 1:** The strongest peak sits at the pore size of 6.0 Å, corresponding well with XRD and the simulation results, although the pores smaller than 4.5 Å cannot be detected by CO₂ gas. Two other peaks below 1 nm in Fig. 2f may imply that some sub-nano graphene capillaries are formed with three or four layers of Ni-pPD nano-islands sandwiched between two

rGO nanosheets. However, because those peaks are relatively weak compared with the main peak at 6.0 Å, it is confident that most nano-islands formed on rGO nanosheets are monolayer Ni-pPD. Also, because the smallest pore size is the determining factor for molecular transport through the macroscopic-scale membranes, the discussion and conclusions made in the main text would not be affected by the pores with larger sub-nano sizes. In addition, the pore size at around 2.5 nm should be considered as the interedge pores between two separated Ni-pPD@rGO nanosheets aligned side by side^[R1].”

5. AFM image in figure 1 – the surface is very smooth whereas with similar height of the flakes supplementary figure 8 is claimed to be rough and that the roughness is due to the monolayer of Ni-pPD on rGO. What is the theoretical molecular size of the Ni-pPD? Does it match with the authors’ claim, “a monolayer of Ni-pPD nano-islands formed on each side of rGO, the repeating unit of the restacked structure between two rGO should be around 1 nm?”

It may be too much of a stretch in analysis to correlate the roughness on some areas of the rGO flakes to be due to the Ni-pPD whereas other areas of the flakes which are equally rough are not considered to be coated with Ni-pPD. For instance, why is the roughness only in the circled area (supplementary Figure 8 reproduced below for convenience) is considered to be coated with Ni-pPD as all through the flake is equally rough.

Reply and revision: Thank you for the referee’s critical comments. Since Ni-pPD is mainly composed of four benzene rings in a plane on top of rGO, the theoretical molecular size of the Ni-pPD nano-islands is similar to that of a monolayer of graphene, 3.4~3.6 Å. This is also confirmed by our DFT calculations in Figure 1e, and by their restacked structures in Figure 2d and Supplementary Figure 8 (previous Supplementary Figure 9).

There are three layers of π - π stacking for the repeating unit of the restacked structure between two rGO. They are, 1) the 1st layer rGO with benzene rings in the 2nd layer Ni-pPD, 2) benzene rings in the 2nd layer Ni-pPD with benzene rings in the 3rd layer Ni-pPD, 3) benzene rings in the 3rd layer Ni-pPD with the 4th layer rGO, resulting in $0.34\sim 0.363\text{ nm} \times 3 = 1.02\sim 1.09\text{ nm}$. Therefore, we claim that “a monolayer of Ni-pPD nano-islands formed on each side of rGO, the repeating unit of the restacked structure between two rGO should be around 1 nm”, and it matches the theoretical molecular size of the Ni-pPD nano-islands.

We do agree with the referee that it may be too much of a stretch in AFM analyses, and thus, we deleted the related discussion and supplementary figure in the revised manuscript.

Thanks to the referee’s valuable suggestion.

References:

[R1] Liu, X. et al. 2D material nanofiltration membranes: from fundamental understandings to rational design. *Adv. Sci.* **8**, 2102493 (2021).

Reviewer #1:

Authors have answered all the queries.

Reviewer #2:

The authors have carefully addressed all the issues I raised previously. I recommend it for publication.

Reviewer #3:

I am satisfied with the revision and can accept the manuscript for publication in its current form.